# An integrated transcriptome mapping the regulatory network of coding and long non-coding RNAs provides a genomics resource in chickpea

Mukesh Jain [1,2✉], Juhi Bansal[1], Mohan Singh Rajkumar[1] & Rohini Garg [3]

Large-scale transcriptome analysis can provide a systems-level understanding of biological processes. To accelerate functional genomic studies in chickpea, we perform a comprehensive transcriptome analysis to generate full-length transcriptome and expression atlas of protein-coding genes (PCGs) and long non-coding RNAs (lncRNAs) from 32 different tissues/organs via deep sequencing. The high-depth RNA-seq dataset reveal expression dynamics and tissue-specificity along with associated biological functions of PCGs and lncRNAs during development. The coexpression network analysis reveal modules associated with a particular tissue or a set of related tissues. The components of transcriptional regulatory networks (TRNs), including transcription factors, their cognate cis-regulatory motifs, and target PCGs/lncRNAs that determine developmental programs of different tissues/organs, are identified. Several candidate tissue-specific and abiotic stress-responsive transcripts associated with quantitative trait loci that determine important agronomic traits are also identified. These results provide an important resource to advance functional/translational genomic and genetic studies during chickpea development and environmental conditions.

[1] School of Computational & Integrative Sciences, Jawaharlal Nehru University, New Delhi 110067, India. [2] National Institute of Plant Genome Research, Aruna Asaf Ali Marg, New Delhi 110067, India. [3] Department of Life Sciences, School of Natural Sciences, Shiv Nadar University, Gautam Buddha Nagar, Uttar Pradesh 201314, India. ✉email: mjain@jnu.ac.in

Chickpea is one of the most important food grain legumes worldwide and is a major source of proteins, carbohydrates and minerals. Despite several efforts via molecular and breeding approaches, chickpea yield has been below its actual potential due to several developmental and environmental constraints[1]. The availability of draft genome sequences of both desi- and kabuli-type chickpeas and pan-genome provides a genetic blueprint for various genomics research in chickpea to enable crop improvement[2–4].

The expression profiles of protein-coding genes (PCGs) and non-coding RNAs in different tissues/organs during plant development can provide important insights into their function and regulation. With the availability of next-generation sequencing technologies, it has been possible to discover novel coding and non-coding genes, and study their gene expression programs with high sensitivity and over a broad range of expression levels[5–8]. Further, transcriptome data can be used to generate co-expression and regulatory networks that can facilitate the prediction of functional roles of an individual or a set of genes and systems-level understanding of biological processes[9–13]. Several large-scale transcriptome analysis studies in different tissues/organs/developmental stages have been undertaken in different legumes via high-throughput sequencing technologies[14–23]. Based on these studies, insights into growth and developmental processes in general or specific processes, such as root nodulation, pollen fertility and seed development in legumes have been obtained[12,14,19,21]. Several RNA-sequencing (RNA-seq) based studies have been performed to measure expression profiles of PCGs in different biological contexts including specific tissues/organs, developmental process and/or environmental stimuli in chickpea[12,13,22,24–28].

Long non-coding RNAs (lncRNAs) represent an emerging regulatory component of the genome that are involved in regulation at transcriptional, post-transcriptional and epigenetic levels. LncRNAs have been implicated in diverse biological processes, such as flowering time, reproductive processes, nutrient response and stress responses[29–32]. A few studies on lncRNAs focused on a particular developmental process and stress conditions have been conducted in chickpea by our group[7,33]. However, a comprehensive analysis of lncRNAs throughout chickpea development is still lacking. Further, transcriptional regulatory network (TRN) analysis including both PCGs and lncRNAs during various developmental processes is also not available. Thus, there is a need for construction of a high-quality full-length transcriptome and generation of a systematic and unified expression atlas of PCGs and lncRNAs to accelerate functional genomic research in chickpea.

To generate a compendium of full-length transcripts and gain insights into the transcriptional regulatory programs underlying different tissues/organs during chickpea development, we generated a comprehensive expression atlas of PCGs and lncRNAs in chickpea. Although short-read sequencing using Illumina is widely used for RNA-seq and is an excellent low-cost technology for quantification of wide-range of gene expression, long-read sequencing can provide more accurate full-length transcripts and their isoforms[27,34]. Thus, we constructed the full-length transcriptome of chickpea using long-read Iso-seq technology of Pacific Biosciences (PacBio) and complemented it with high-depth RNA-seq from Illumina sequencing. We described dynamic transcriptional profiles, tissue-specific expression and co-expressed modules of PCGs and lncRNAs along with their functional relevance. In addition, we elucidated the putative TRNs operative in different tissues/organs during development in chickpea. Further, we revealed the candidate tissue-specific and abiotic stress-responsive transcripts associated with quantitative trait loci (QTLs) for important agronomic traits. The high-resolution transcriptional map reported here provides a comprehensive resource for prioritization of candidate PCGs and lncRNAs for functional analysis and crop improvement.

## Results

**Generation of full-length reference transcriptome assembly**. We followed a comprehensive strategy to generate a full-length transcriptome assembly for chickpea using high-depth long-read PacBio and short-read Illumina sequencing data (Supplementary Fig. 1). The PacBio sequencing of four different size-selected cDNA libraries (Iso-seq; Supplementary Table 1) generated a total of 81,269 non-redundant transcript isoforms with an average length of 2631 bp (Supplementary Table 2 and Supplementary Fig. 1). The reference-based assembly of ~6.5 billion high-quality reads (from 94 libraries representing 32 different tissues/organs) obtained using the Illumina platform (Fig. 1a and Supplementary Data 1), resulted in a total of 55,313 unique transcripts with an average length of 1980 bp (Supplementary Table 2). Overall, a higher fraction (~40%) of transcripts obtained via Iso-seq were longer as compared to that of transcripts obtained from the assembly of Illumina RNA-seq data (Fig. 1b and Supplementary Data 2). The average length and N50 length of transcripts obtained via Iso-seq were also greater (Supplementary Table 2). Further, transcripts obtained from PacBio and Illumina platforms were merged and clustered into 111,620 transcripts (Supplementary Table 2). These transcripts represented isoforms with intron retention (IR), exon skipping (ES), alternate acceptor (AA), and alternate donor (AD) as major alternative splicing (AS) events. Among them, the IR type AS event was most frequent (Fig. 1c). The biological process GO terms, such as protein modification process, ion/protein transport, root and shoot system development, reproductive development and small RNA mediated processes were enriched in all types of AS events (Supplementary Fig. 2). However, the biological processes related to coenzyme metabolism, response to biotic/abiotic stress, carbohydrate mediated signaling/metabolism, cell cycle, differentiation and regulation of nucleic acid were largely enriched in the transcript isoforms with IR type AS event. Likewise, mRNA metabolic processes were exclusively enriched in the transcripts showing ES events (Supplementary Fig. 2).

The 111,620 transcript isoforms represented a total of 38,818 unique transcripts/gene loci. The final reference transcriptome assembly was of 87.69 Mb size with an average and N50 transcript lengths of 2259 and 3292 bp, respectively (Supplementary Table 2). The reference transcriptome assembly reported here showed greater contiguity and completeness (Supplementary Table 3). At least 35,215 transcripts aligned with the chickpea genome with high coverage (≥95% identity and ≥70% coverage). The list of 3603 transcripts that did not align with the genome is provided in Supplementary Data 3. Likewise, of the 28,269 PCGs predicted in the chickpea genome[3], at least 24,581 PCGs were represented in the reference transcriptome assembly (with at least 95% identity and 30% coverage) (Supplementary Fig. 3). The remaining PCGs predicted in the chickpea genome may represent the annotation artifacts or these genes might be expressed in highly specific cell types and are not represented in our transcriptome assembly. Overall, we identified a total of 13,586 unique transcripts in the reference transcriptome assembly that were not predicted earlier in the chickpea genome annotation (Supplementary Data 3).

**Identification and conservation of lncRNAs**. A total of 5293 unique transcripts were identified as high-confidence lncRNAs using reference transcriptome assembly. The remaining 33,525 transcripts represented PCGs. Both PCGs and lncRNAs were

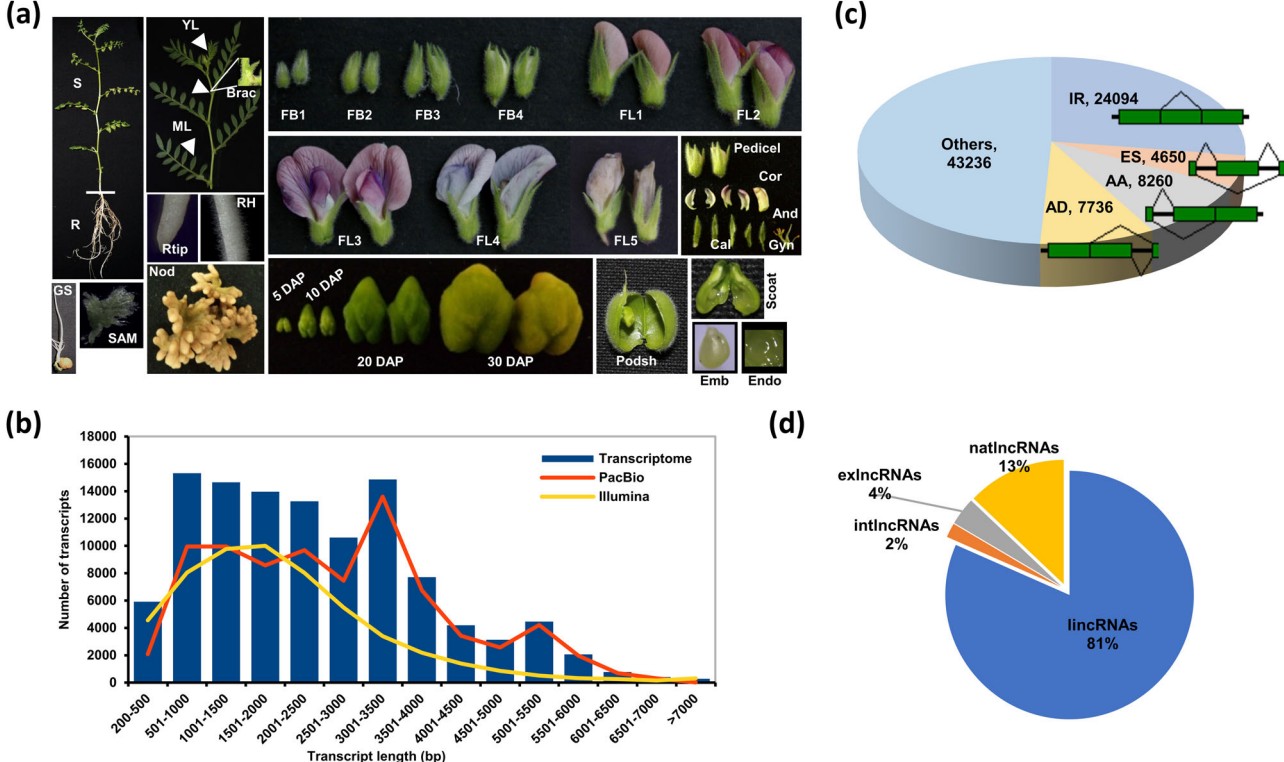

**Fig. 1 Sequencing and analyses of chickpea full-length transcriptome. a** Different tissues/organs used for generation of RNA-sequencing data using Illumina platform. GS germinating seedling, S shoot, ML mature leaf, YL young leaf, Brac bracteole, R root, Rtip root tip, RH root hair, Nod nodule, SAM shoot apical meristem, FB1–FB4 stages of flower bud development, FL1–FL5 stage of flower development, Cal calyx, Cor corolla, And androecium, Gyn gynoecium, Pedi pedicel, Emb embryo, Endo endosperm, SdCt seed coat, PodSh podshell, 5DAP seed 5 days after pollination, 10DAP seed 10 days after pollination, 20DAP seed 20 days after pollination, 30DAP seed 30 days after pollination. **b** Length distribution of transcripts in the final transcriptome assembly and the transcripts generated via PacBio sequencing and reference-based assembly of Illumina data alone from 32 tissue samples. **c** Pie-chart showing the fraction of different types of major alternative spliced events (IR intron retention, ES exon skipping, AA alternate acceptor, AD alternate donor) represented in the transcriptome. **d** Classification of predicted lncRNAs based on their genomic location. lincRNAs intergenic lncRNAs, natlncRNAs natural antisense lncRNAs, exlncRNAs exonic lncRNAs, intlncRNAs intronic lncRNAs.

found to be uniformly distributed on all the chickpea chromosomes (Supplementary Fig. 4a and Supplementary Data 2). A comparative analysis of lncRNAs and PCGs revealed an average shorter length, higher AU nucleotide content and fewer exons in lncRNAs as compared to the PCGs (Supplementary Figs. 4b and 5a and Supplementary Data 2). Of the 4775 lncRNAs mapped to the chickpea genome, 3123 (65.4 %) had a single exon and the remaining lncRNAs harbored two or more exons (Supplementary Fig. 5b and Supplementary Data 2). The lncRNAs were classified into different classes based on their genomic location (Fig. 1d). The largest number of lncRNAs were classified as intergenic (lincRNAs; 81%), followed by natural antisense (natlncRNAs; 13%), exonic (exlncRNAs; 4%) and intronic (intlncRNAs; 2%).

Only 5.4% (280) of the identified chickpea lncRNAs showed significant similarity with the available (~0.4 million) lncRNA sequences from 60 different plant species in GREENC and CANTATAdb databases. The maximum conservation was observed with plant species belonging to the Fabaceae family, the largest being with *Medicago truncatula* (230) followed by *Trifolium pratense* (136) and *Glycine max* (123; Supplementary Table 4). Only 111 lncRNAs showed conservation in at least two species (other than chickpea) belonging to the Fabaceae family followed by 11 each in Poaceae and Brassicaceae families (Supplementary Fig. 6a). Further, only 22–31% of the lncRNAs showed significant similarity with the previously reported set of lncRNAs in chickpea[7,33]. These results indicate very low

conservation of the lncRNAs as shown in the earlier studies too[7,30,32,33].

**Functional annotation of PCGs and lncRNAs.** A total of 32,708 (97.6%) PCGs could be annotated using different databases. Most of the novel PCGs identified in this study showed significant similarity with predicted genes/transcripts in other plant species. The largest fraction of PCGs belonged to the regulation of transcription (4.9%) and defense response (4.1%) biological process gene ontology (GO) terms, and ATP binding (18.2%) and metal ion binding (11.7%) molecular function GO terms (Supplementary Fig. 7a and Supplementary Data 2). The protein kinase followed by pentatricopeptide repeat represented the most frequent Pfam domains among the PCGs (Supplementary Fig. 7b and Supplementary Data 2). The novel transcripts showed enrichment of the GO terms related to development, stress and metabolism/ cellular processes (Supplementary Fig. 8 and Supplementary Data 2). Next, we identified at least 2629 (7.84%) PCGs encoding for transcription factors (TFs) belonging to 85 different families. The AP2-EREBP (180), MYB (143), bHLH (131), HB (125), and CCHC (106) families were most highly represented among the identified TFs (Supplementary Fig. 9a, Supplementary Table 5, and Supplementary Data 2). A similar distribution of genes in different TF families with little variations was observed in soybean and model plant Arabidopsis too (Supplementary Fig. 9b, Supplementary Table 5 and Supplementary Data 2).

The lncRNAs represent a part of the non-coding component of the genome and do not show conservation across plant species. Therefore, the functional annotation of lncRNAs was performed based on their putative target PCGs. It has been demonstrated that many lncRNAs act in *cis* to regulate the expression of their proximal PCGs[30,32,35]. We identified a total of 4288 *cis*-target PCGs based on their proximity with one or more lncRNAs and 2100 *trans*-target PCGs based on the correlation of their expression patterns (Supplementary Data 4). Based on GO annotation of their putative target PCGs, 4882 lncRNAs could be assigned with at least one GO term. The largest fraction of lncRNAs were found associated with regulation of transcription (7.9%) and protein phosphorylation (6.4%) biological process GO terms (Supplementary Figs. 6b and 10). Likewise, ATP binding (20.8%), metal ion binding (14.1%) and transcription factor activity (11%) were the most represented molecular function GO terms among the lncRNAs (Supplementary Fig. 10 and Supplementary Data 2). Further analysis revealed significant enrichment of lncRNAs associated with signaling, root and shoot system development, reproductive development processes, seed development, response to stress and hormones, and metabolic processes (Supplementary Figs. 6b and 10 and Supplementary Data 2), indicating their role in diverse cellular and biological processes. The complete functional annotation of PCGs and lncRNAs is available in Supplementary Data 4.

**Expression dynamics across different tissues/organs in chickpea**. We profiled 32 different tissues/organs spanning vegetative and reproductive stages of development in chickpea (Fig. 1a) to generate a comprehensive integrated expression atlas of PCGs and lncRNAs. Ten samples (GS, germinating seedling; S, shoot; ML, mature leaf; YL, young leaf; Bract, bracteole; R, root; Rtip, root tip; RH, root hair; Nod, nodule and SAM, shoot apical meristem) represented vegetative tissues/organs and 22 samples (FB1-FB4, stages of flower bud development; FL1-FL5, stage of flower development; Cal, calyx; Cor, corolla; And, androecium; Gyn, gynoecium; Pedi, pedicel; Emb, embryo; Endo, endosperm; SdCt, seed coat; PodSh, podshell; 5DAP, seed 5 days after pollination; 10DAP, 20DAP and 30DAP) represented tissues/organs from various stages of reproductive development. The biological replicates of different tissues/organs showed high reproducibility with Pearson correlations (r) among them ranging from 0.88 to 0.99 (average of 0.96; Supplementary Figs. 11 and 12). Overall, a total of 36,600 (94.3%) transcripts (31,616 PCGs and 4984 lncRNAs) were found to be expressed (FPKM ≥0.5) in at least one of the tissues/organs analyzed (Supplementary Data 5). The percentage of total transcripts expressed in different tissues/organs ranged from 57.8% in 30DAP to 74.2% in 20DAP tissue (average of 69%). The number of non-TF PCGs, TF-encoding genes and lncRNAs were expressed proportionately in different tissues/organs (Fig. 2a and Supplementary Data 2). A large fraction of the lncRNAs were expressed at lower level as compared to the PCGs in most of the tissues/organs analyzed (Supplementary Fig. 13), which is consistent with previous studies in chickpea and other plants[7,30,32,33].

The transcriptome dynamics and relationships among the transcriptomes of different tissue samples were explored via the non-metric multidimensional scaling (NMDS) method and hierarchical clustering based on Pearson's correlations. Most of the samples with similar developmental origins tended to be grouped closer (Fig. 2b, Supplementary Figs. 11 and 12). The root and root-derived tissues (root tip, root hair, and nodule) grouped closely and were labeled as root tissues (RT). The tissue representing different stages of flower development including FB1–FB4 and FL1–FL5 were clustered together as flower

development stages (FDS). Likewise, stages of seed development (5DAP–30DAP) and seed/pod tissues (embryo, endosperm and seed coat), were grouped together as seed tissues (ST). However, podshell showed a higher correlation with the green tissues. The green leaf-like tissues, including shoot, bracteole, mature leaf and young leaf were grouped into green tissues (GT). Among the flower parts (FP), corolla, androecium and gynoecium showed somewhat similar transcriptomes. However, pedicel and calyx being leaf-like structures were more correlated with GT.

**Tissue specificity of PCGs and lncRNAs, and their functional relevance**. The tissue-specific PCGs and lncRNAs represent important components involved in the developmental processes of respective tissue/organ. We identified a total of 5611 (16.7%) PCGs including 371 TF-encoding genes as tissue-specific (Supplementary Data 6). The highest number of tissue-specific PCGs were detected at 20DAP seed stage (2103) followed by nodule (736), endosperm (689), and androecium (372) (Fig. 3a). A considerably higher proportion of lncRNAs (2290; 43.3%) was found expressed in a tissue-specific manner; largest being in endosperm (430), 20DAP (369) followed by androecium (279), 5DAP (255), and nodule (255) tissues (Fig. 3b). Overall, a substantial proportion (14.9–72.7%) of tissue-specific transcripts were represented by lncRNAs in each tissue/organ (Supplementary Fig. 14 and Supplementary Data 2).

*Functional relevance of tissue-specific PCGs.* The comparative GO enrichment map analysis of the sets of tissue-specific PCGs in the five groups (GT, RT, FDS, FP and ST) of tissues/organs (Fig. 3c) highlighted the enrichment of GO terms, including flavonoid/terpenoid metabolic processes, hormone metabolic processes, lipid metabolic process, cell wall organization, stem cell division, response to jasmonic/salicylic acid and reactive oxygen species in the RT group tissues. The gametophyte development processes and fatty acyl Co-A metabolic processes were enriched in the FDS group of tissues. The GO terms related to defense response, programmed cell death, male gametogenesis, response to amino acids and ion transport/homeostasis were enriched in FP. The PCGs associated with reproductive process-related terms were found enriched in the ST group. Further, the GO enrichment analysis of individual sets of tissue-specific PCGs revealed the enrichment of biological processes relevant to specific tissues/organs in different groups (Supplementary Fig. 15). For example, the GO terms associated with growth and defense, stress response, response to ethylene, lignin biosynthesis, auxin transport, and cell wall organization were found enriched in root, root tip, and/or root hair tissues. The GO terms, defense/modulation by symbiont of host defense response and plant-type hypersensitive response were associated with genes expressed specifically in the nodule. A higher number of genes involved in gametophyte development, microgametogenesis and cellular carbohydrate metabolic processes, were found to be specifically expressed/enriched in FDS and FP tissues. The GO terms, like cell wall modification, cell fate determination, auxin and gibberellin mediated signaling, and regulation of DNA methylation were found to be significantly enriched in endosperm. Many GO terms associated with seed development processes, such as lipopolysaccharide biosynthetic process, auxin polar transport, fatty acid biosynthetic process, gluconeogenesis, acquisition of desiccation tolerance and dormancy process were significantly enriched in 5DAP–30DAP seeds (Supplementary Fig. 15).

Further, a large fraction (371, 14.1%) of TF-encoding genes exhibited tissue-specific expression (Supplementary Fig. 16). The largest number of TF-encoding genes were expressed specifically at 20DAP stage (149) followed by endosperm (63) among the

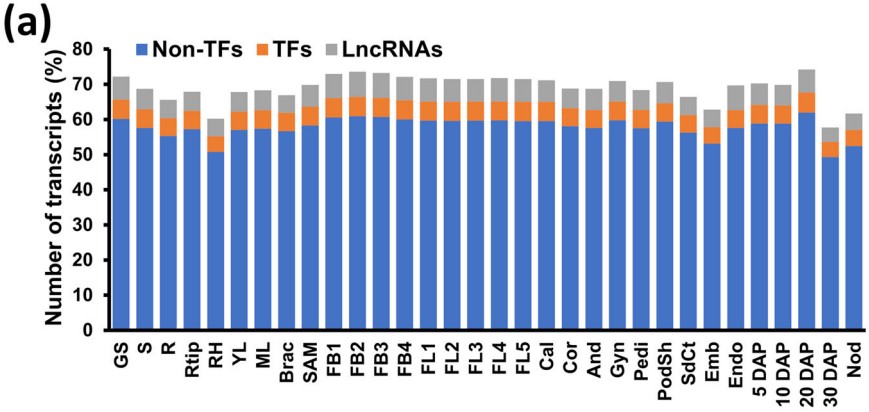

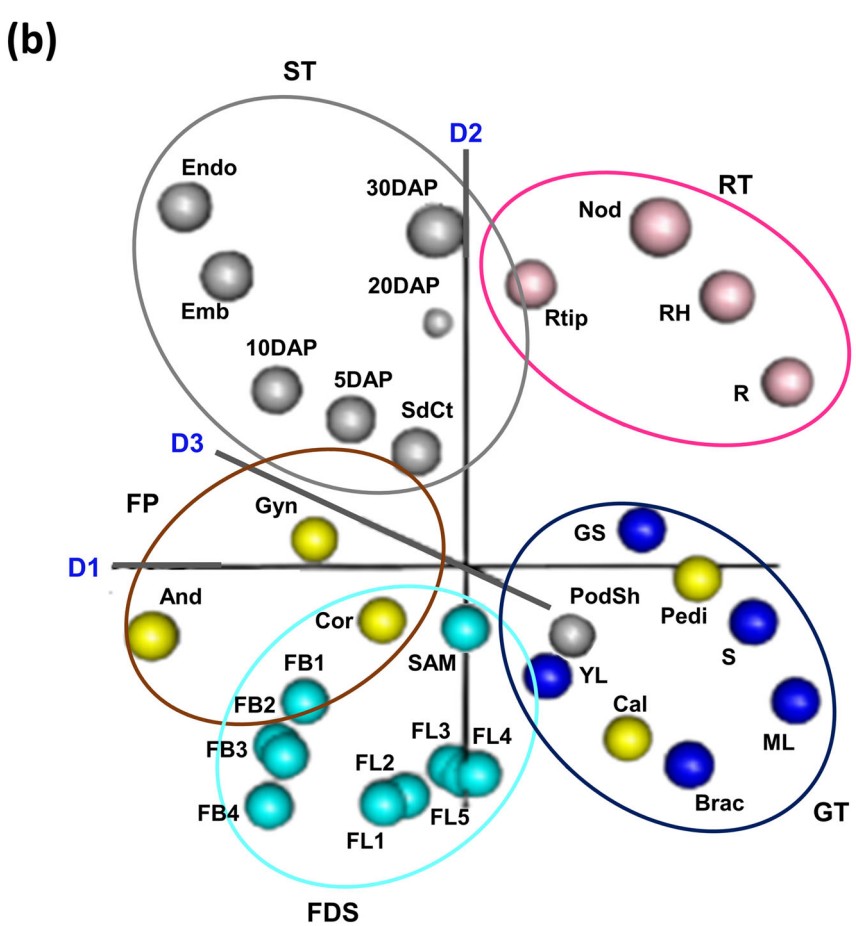

**Fig. 2 Expression patterns and correlation of transcriptome in different tissues/organs in chickpea. a** Percentage of transcription factor (TF) encoding genes, non-TF protein-coding genes (PCGs) and lncRNAs expressed in the 32 tissues/organs analyzed. **b** Non-metric multidimensional scaling analysis (NMDS) plot showing correlation among the transcriptomes of 32 tissue/organs samples. The tissues/organs were grouped broadly in five groups; Green tissues (GT, blue), Root tissues (RT, pink), Flower development stages (FDS, cyan), Flower parts (FP, orange), and Seed tissues (ST, gray), indicated in different colors. GS germinating seedling, S shoot, ML mature leaf, YL young leaf, Brac bracteole, R root, Rtip root tip, RH root hair, Nod nodule, SAM shoot apical meristem, FB1–FB4 stages of flower bud development, FL1–FL5 stage of flower development, Cal calyx, Cor corolla, And androecium, Gyn gynoecium, Pedi pedicel, Emb embryo, Endo endosperm, SdCt seed coat, PodSh podshell, 5DAP seed 5 days after pollination, 10DAP seed 10 days after pollination, 20DAP seed 20 days after pollination, 30DAP seed 30 days after pollination. D1, D2, and D3 represent first, second, and third dimension, respectively.

seed tissues. Among the root tissues, the highest number of TF-encoding genes exhibited tissue-specific expression in root hair (22) and root tip (17), and members of bHLH followed by MYB, AP2-EREBP, and LOB family TFs were found to be most represented. Likewise, AP2-EREBP, C2H2, CCHC, and/or CCAAT TFs displayed tissue-specific gene expression among the green tissues and flower parts. LOB and NAC TFs displayed

tissue-specific expression during flower development stages (Supplementary Fig. 16).

*Functional relevance of tissue-specific lncRNAs.* The comparative GO enrichment map analysis of the sets of lncRNAs specifically expressed in the five groups of tissues/organs and GO enrichment analysis of individual sets of tissue-specific lncRNAs suggested

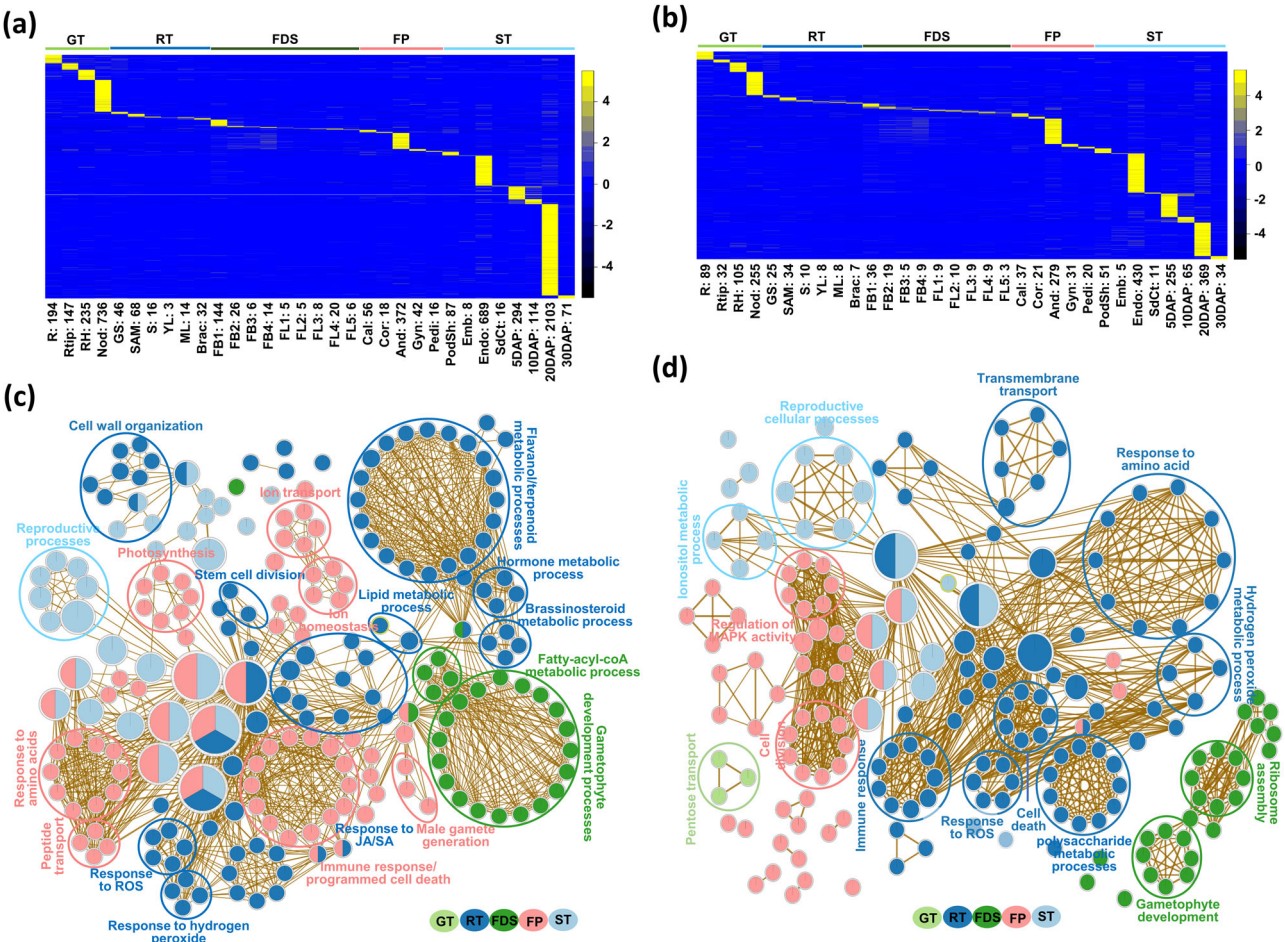

**Fig. 3 Tissue-specificity and comparative gene ontology (GO) enrichment map analysis. a, b** Heatmaps showing expression profiles of the tissue-specific protein-coding genes (PCGs) (**a**) and lncRNAs (**b**) in different tissue samples. Numbers given at the bottom indicate the number of tissue-specific genes/lncRNAs identified in each tissue sample. Color scales showing the Z-score are given on the right side. **c, d** Comparative GO enrichment maps of the tissue-specific PCGs (**c**) and lncRNAs (**d**) in the five groups. GO terms associated with different groups, green tissues (GT), root tissues (RT), flower development stages (FDS), flower parts (FP), and seed tissues (ST), are indicated in different colors. Only the significantly enriched (P value ≤0.05) biological process GO terms are shown.

their roles in several important biological processes (Fig. 3d and Supplementary Fig. 17). The lncRNAs associated with GO terms, including transmembrane transport, cell death, polysaccharide metabolic processes, immune response, auxin stimulus, defense response, hexose catabolic process, root hair initiation/differentiation, response to ROS, and hydrogen peroxide metabolic process were found enriched specifically in the RT group of tissues (Supplementary Fig. 17). A few lncRNAs associated with cell wall biogenesis, carbohydrate biosynthetic process, glycoprotein biosynthetic process, gametophyte development and response to cytokinin stimulus were found enriched in the FDS tissues. DNA demethylation, regulation of proteolysis, and stomatal complex morphogenesis terms were associated with SAM-specific lncRNAs. The GO terms, including amino acid transport, floral organ development, lipid catabolic processes, pollen maturation, cell cycle, chromatin silencing and carbohydrate metabolic processes, were associated with lncRNAs specifically expressed in different floral organs. In ST group, the reproductive cellular process and inositol metabolic process were found enriched among the associated lncRNAs. The lncRNAs associated with regulation of transcription, gene silencing by miRNA, flavonoid biosynthetic process and cytokinin-mediated signaling were enriched in endosperm. The GO terms, including regulation of cell size, cell differentiation, xyloglucan metabolic process,

transport, histone lysine methylation, lipid biosynthetic process, and seed oil body biogenesis were associated with stages of seed development (Supplementary Fig. 17).

**Co-expression network and co-expressed modules**. We performed weighted gene coexpression network analysis (WGCNA) to determine the sets of coexpressed PCGs and lncRNAs in different tissues/organs. The WGCNA analysis of all the transcripts showing a variance of >1.5 across different tissues/organs identified a total of 24 modules (named as M1 to M24 hereafter) representing the sets of transcripts with similar expression patterns (Supplementary Fig. 18, Fig. 4a and Supplementary Data 2). The number of transcripts in different modules ranged from 64 in M3 and M8 to 5774 in M2 (Supplementary Data 7). All the modules harbored TF-encoding genes, non-TF PCGs and lncRNAs (Fig. 4a and Supplementary Data 2). Most of the modules displayed a significant correlation (≥0.70) with one or more of the related tissues/organs (Fig. 4b). For example, module M1 was found significantly correlated ($r = 0.71$; $P = 5e{-}06$) with embryo, whereas M6 ($r = 0.89$; $P = 7e{-}12$) and M7 ($r = 0.71$; $P = 5e{-}04$) modules correlated significantly with endosperm. M2 ($r = 0.75$; $P = 1e{-}05$) and M3 ($r = 0.70$; $P = 8e{-}05$) modules showed high correlation with 20DAP and M5 module with

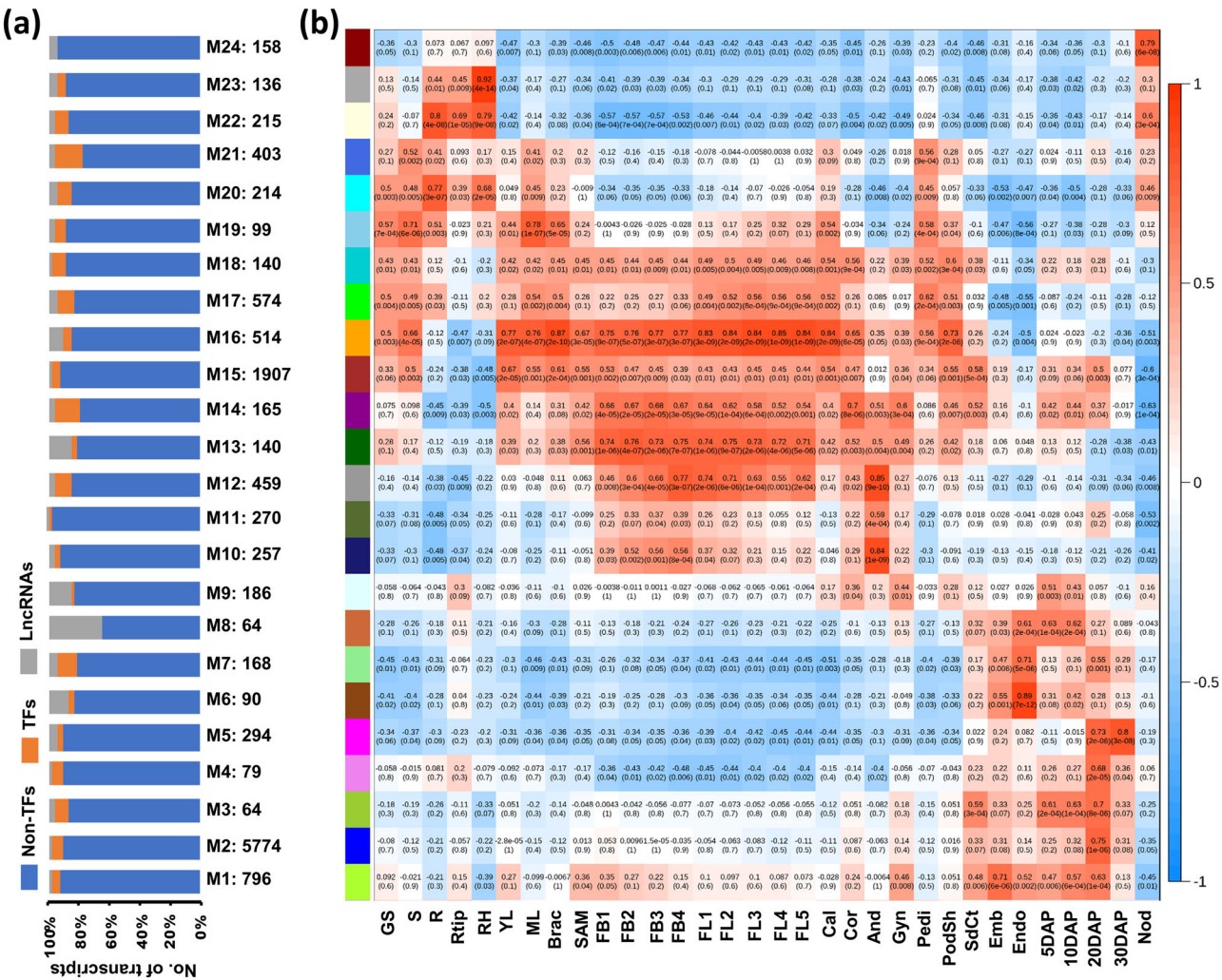

**Fig. 4 Coexpression network analysis in chickpea. a** Bar graph showing the fraction of transcription factor (TF) encoding genes, non-TF protein-coding genes and lncRNAs included in each module (M1–M24). The total number of transcripts included in each module are given below each bar. **b** Heatmap showing correlation of each module (M1–M24) with different tissue samples. The correlation value and significance value are given in each box. The correlation values are shown in color scale also.

20DAP ($r = 0.73$; $P = 2e-06$) and 30DAP ($r = 0.80$; $P = 3e-05$) stages of seed development. M10 and M12 modules exhibited the highest correlation with androecium. M12 and M13 were correlated ($r = 0.71-0.76$; $P \leq 1e-04$) with stages of flower bud and flower development. M20 and M22 showed higher correlation ($r = 0.77-0.80$; $P \leq 1e-04$) with root tissues, including root, root tip, and root hair. M24 was correlated specifically with nodule ($r = 0.79$; $P \leq 6e-08$) and M23 ($r = 0.92$; $P \leq 4e-4$) with root hair. The coexpressed transcripts in each module highlighted the characteristic biological processes related to the particular tissue/organ in most cases.

**Transcriptional regulatory networks (TRNs) involved in developmental processes**. We analyzed TRNs associated with developmental processes via analyzing the co-expressed module(s) in individual or a set of related tissues/organs (Figs. 5 and 6). In this analysis, we analyzed co-expressed transcripts within a single or set of modules and associated the TFs with their putative target PCGs and/or lncRNAs harboring their binding sites in their upstream regions followed by assignment of GO terms to the corresponding set of transcripts.

Several *cis*-regulatory motifs were found enriched in the promoters of both PCGs (39 TFs and 370 non-TF PCGs) and

lncRNAs (20) included in the modules M20 and M22, that were correlated with root and/or related tissues (root tip and root hair) (Fig. 5a). Among these, binding motifs of MYB (MYB13/49/81/55 and BOS1), C2H2 (G26030), WRKY (WRKY18), bZIP (TGA10), and NAC (ATAF1) TFs were most represented. A large fraction of target PCGs and lncRNAs belonged to stress responses, metabolic processes, cell differentiation and regulation of root development. In addition, PCGs associated with hormone signaling and response, transport (sucrose and amino acid), meristem growth, root hair elongation, nodulation and root development, were also represented in this set of coexpressed transcripts. The representation of several enriched *cis*-regulatory motifs and biological processes related to the root development among lncRNAs indicate their role in development of root tissues.

Two modules, M12 and M13, were associated with most stages of flower bud and flower development (Fig. 5b). This set of coexpressed transcripts harbored *cis*-regulatory motifs, ATAF1/ANAC38/ANAC70, SOX2/15, bZIP, At5g61620, AtHB23/24, SEP3, NLP7, and MYB56/70/83, representing the binding sites of different TF families. The target PCGs included in these modules were found to be involved in several characteristic developmental processes of flower development, such as floral

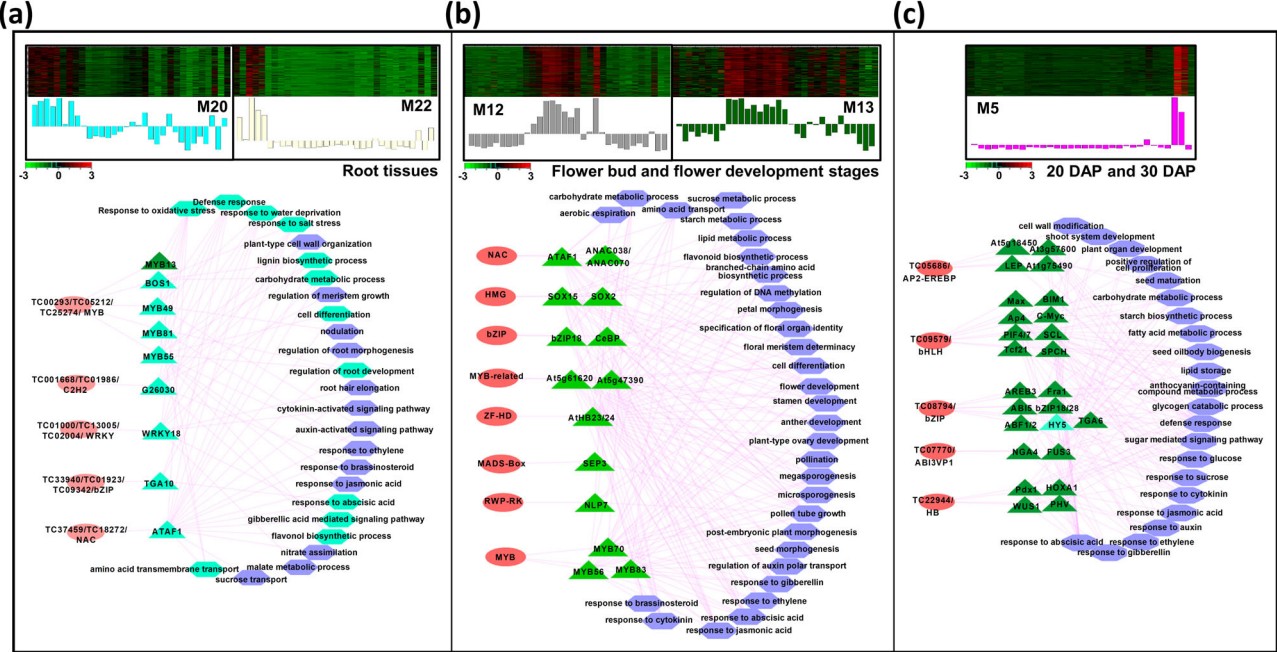

**Fig. 5 Expression profile and transcriptional regulatory network associated with modules correlated to multiple tissues or stages of development.** Heatmaps show the expression profile of all the coexpressed genes in the module(s) correlated with multiple tissues, including root tissues (**a**), flower bud and flower development stages (**b**), and 20DAP and 30DAP stages of seed development (**c**). Color scales represent *Z*-score. Bar graphs (below the heat maps) show the consensus expression pattern of the coexpressed transcripts in each module. The predicted transcriptional regulatory network (significantly enriched TF-binding sites along with the associated TFs and GO terms) associated with the transcript sets in the modules are shown below the heatmaps. The significantly enriched cis-regulatory motifs only in protein-coding genes (PCGs), and both PCGs and lncRNAs are shown by green and sea green triangles, respectively. The associated GO terms only in PCGs, and both PCGs and lncRNAs are shown by blue and sea green hexagons, respectively. The TFs are represented by pink ovals. Edges represent known interactions between the cis-regulatory motifs and transcription factors.

organ specification, flower development, stamen/anther development, pollination, micro/megasporogenesis and petal morphogenesis. In addition, genes associated with sucrose, carbohydrate, starch and lipid metabolic processes, and various hormonal responses were also found in this set of transcripts.

The transcripts in the M5 module with higher expression at 20DAP and 30DAP stages of seed development showed enrichment of *cis*-regulatory motifs, LEP, PIF4/7, AREB, SCL, ABI5, bZIP18/28, FUS3, WUS1, PHV and HOX, which represent the target sites of AP2-ERBP, bHLH, bZIP, ABI3VP1 and HB family TFs (Fig. 5c). The target transcripts were found implicated in cell wall modification, starch biosynthetic process, lipid storage, seed oil body biogenesis, defense response, and response to different plant hormones along with sugar-mediated signaling pathway.

The transcripts in the M23 module correlated with root hair were enriched in ATY19, MYB and WRKY *cis*-regulatory motifs representing the binding sites of MYB and WRKY TFs, and were associated with biological processes, such as root hair elongation, cell differentiation, regulation of cell division, cell wall biogenesis/ modification, carbohydrate/lipid metabolic processes, auxin-activated signaling, and response to stress (salt stress, oxidative stress, and drought recovery) (Fig. 6a).

In androecium-specific modules (M10 and M12), *cis*-regulatory motifs like NLP7, SCL, SOX2/15, bZIP18, ANL2, PHV, MYB70, and KANAD11 representing the binding sites of RWP-RK, bHLH, MYB, MYB-related, HMG, and bZIP TFs were significantly enriched (Fig. 6b). These modules included the target genes involved in several processes directly related to androecium development, such as anther/stamen/pollen development, pollen differentiation, pollen wall assembly and pollen tube growth. In addition, other biological processes, including cell division, auxin

transport, lignin and lipid metabolic processes, carbohydrate metabolic process, fatty acid biosynthesis, and various hormonal responses were also represented among the target genes in these modules.

The M3 and M4 modules correlated with 20DAP stage of seed development, were found to be enriched in a large number of *cis*-regulatory motifs, including ARF, ANAC, CUC1-3, AGL, ERF, bHLH, TGA, ABI, ABF, KNOTTED, WUS, etc., which represent the binding sites of several TFs, such as ARF, NACs, MADS-box, AP2-EREBP, bZIP, HB and WRKY (Fig. 6c). Most of the enriched *cis*-regulatory motifs were detected in both PCGs and lncRNAs. GO terms, such as seed coat development, post-embryonic development and cell differentiation were found to be represented among the target genes. The representation of GO terms, such as cell proliferation, regulation of cell growth, cell wall organization, glycogen biosynthetic process and oil body biogenesis was concurrent to the developmental processes occurring inside the seed at 20DAP stage. The enrichment of regulation of DNA endoreduplication GO term at this stage is in concordance with the grain filling process that occurs during the mid-maturation stage of seed development, which might determine seed size[10]. Further, genes involved in epigenetic processes, such as maintenance of chromatin silencing, regulation of gene expression via genetic imprinting, siRNAs involved in RNA interference and gene silencing via miRNAs were also detected, indicating importance of these processes during seed development.

M1 module correlated with embryo and showed enrichment of *cis*-regulatory motifs, SCL, SOX2/4, HDG1, ARF2, ERF, MYB, LHY, and GATA that represent the binding sites of bHLH, HB, AP2-EREBP, MYB, and C2C2-GATA family TFs (Fig. 6d). The transcripts harboring these motifs were found associated with GO

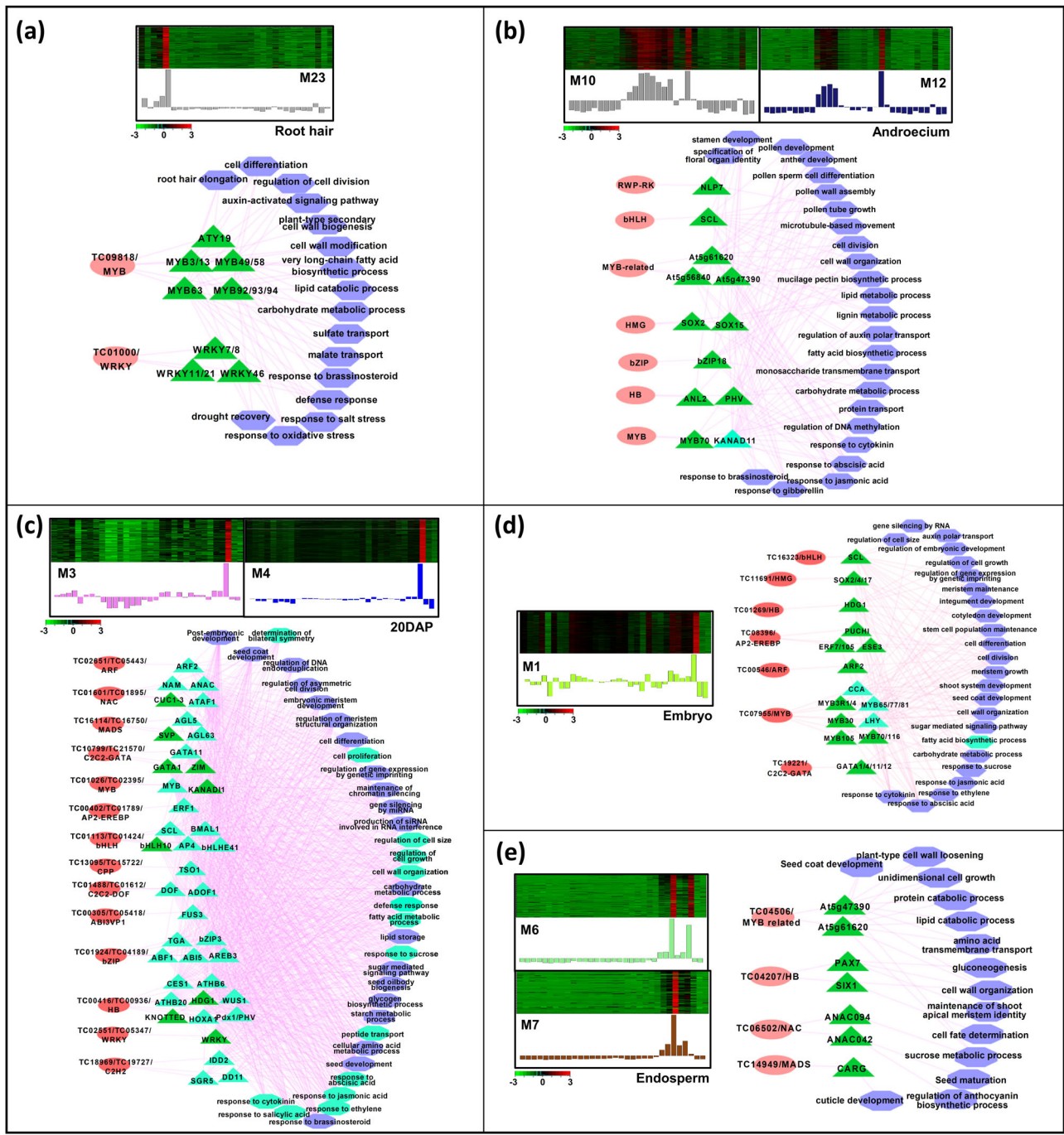

**Fig. 6 Expression profile and transcriptional regulatory network associated with the modules correlated to a specific tissue or stage of development.**
Heatmaps show the expression profile of all the coexpressed genes in the module(s) correlated with the given specific tissue, including root hair (**a**), androecium (**b**), 20DAP (**c**), embryo (**d**), and endosperm (**e**). Color scales represent Z-score. Bar graphs (below the heat maps) show the consensus expression pattern of the coexpressed transcripts in each module. The predicted transcriptional regulatory network (significantly enriched TF-binding sites along with the associated TFs and GO terms) associated with the transcripts sets in the modules are shown below the heatmaps. The significantly enriched cis-regulatory motifs only in protein-coding genes (PCGs), and both PCGs and lncRNAs are shown by green and sea green triangles, respectively. The associated GO terms only in PCGs, and both PCGs and lncRNAs are shown by blue and sea green hexagons, respectively. The TFs are represented by pink ovals. Edges represent known interactions between the cis-regulatory motifs and transcription factors.

terms related to cell division, growth and differentiation, meristem growth/maintenance, fatty acid/carbohydrate metabolic processes, sugar signaling and response to various plant hormones. In addition, the terms related to epigenetic regulation, such as gene silencing by RNA and genetic imprinting were also represented among the target genes included in this module. The endosperm associated modules (M6 and M7) were found to

harbor significantly enriched motifs, including At5g47390 and At5g61260, PAX7, SIX1, ANAC094/042, and CARG (Fig. 6e). These motifs represented the binding sites of MYB-related, homeobox, NAC, and/or MADS-box TFs. The target genes were associated with GO terms, including seed coat development, cell wall organization/loosening, protein and lipid catabolic processes, gluconeogenesis, and cell fate determination. Many of these

components and biological processes have been found associated with embryo and endosperm in soybean too[36].

**QTL-associated transcripts showing tissue-specific expression.**
To identify the candidate transcripts that may determine important agronomic traits, we integrated the tissue-specific expression with the known QTLs associated with seed size/weight (100 seed weight, 100 SDW) and other traits identified in previous studies[37–44]. A total of 74 transcripts located within the 100 SDW associated QTLs were identified (Supplementary Data 8). Interestingly, all of these transcripts displayed specific expression during the stages of seed development, including 65 transcripts with specific expression at 20DAP stage, and four each at 5DAP and 30DAP stages, and one at 10DAP stage (Fig. 7a and Supplementary Data 8). Of these, four transcripts (TC01465, TC11584, TC17537, and TC23766) represented lncRNAs and eight transcripts encoded TFs belonging to different families (TC00497, AP2-EREBP; TC00082, C2C2-GATA; TC00961, HB; TC14173, bZIP; TC23573, bHLH; TC36066, FHA; TC23790, SET; and TC26366, NAC). At least 23 transcripts were associated with coexpression modules mostly with M2 and M5 that are correlated with 20DAP stage of seed development (Fig. 7a). These transcripts were found distributed on the chickpea genome; largest number being on chromosome 1 (19) followed by chromosome 4 (15), 8 (13), and 7 (12) (Fig. 7b). Among the candidate transcripts located on chromosome 4, at least six were found associated with the QTL-hotspot region (Fig. 7b). These transcripts were found implicated in transcription factor/regulatory activity, stress response, carbohydrate and lipid metabolic process, regulation of cell size, generation of precursor metabolites and energy, carbohydrate and lipid metabolic process, catalytic activity and transporter activity molecular functions (Supplementary Data 8).

Next, 16 PCGs located within QTLs associated with other agronomic traits [11 PCGs in harvest-index (HI) QTLs, one in plant height (PHT) QTL, and four in QTLs associated with multiple traits] also exhibited tissue-specific expression (Fig. 7a and Supplementary Data 8). Six PCGs (TC18138 encoding hydroxysteroid dehydrogenase 1, TC36990 encoding CLPC homolog 1, TC18173 encoding homeodomain-like superfamily protein, TC18186 encoding ferrochelatase 2, TC36933 encoding CBS domain-containing protein and TC18252 encoding an integral component of membrane) located in HI QTLs were expressed specifically at the 20DAP stage and one gene at each of 10DAP (TC18263 encoding gibberellin 20 oxidase) and 30DAP (TC18135 encoding hydroxysteroid dehydrogenase 1) stages (Fig. 7a, b). The three remaining HI QTL associated PCGs showed specific expression one each in nodule (TC18107 encoding remorin family protein), root (TC34119 encoding SUMO-activating enzyme 1A) and SAM (TC32915 encoding a glycosyl-transferase) (Fig. 7a, b). All the HI QTL associated PCGs were found clustered together at chromosome 6 (Fig. 7b). One PHT QTL associated PCG (TC17824 encoding a sugar transporter) at chromosome 6 was expressed specifically at the 20DAP stage of seed development (Fig. 7a, b). Further, three PCGs, two at 20DAP stage of seed development (TC35237 encoding an N-acetyltransferase and TC37279 encoding a cupin family protein) and one at FL4 stage of flower development (TC38316 encoding homocysteine S-methyltransferase 3) located within overlapping R-T ratio (RTR), shoot dry weight (SDW) and 100 SDW QTLs exhibited specific expression. Another gene (TC36647 encoding a protein serine/threonine kinase) located within overlapping QTLs associated with root length density (RLD), PHT, days to maturity (DM), pods per plant (POD), 100 SDW and delta carbon ratio (DC) traits exhibited specific expression in the nodules (Fig. 7a, b). These four candidate PCGs associated with multiple QTLs were located within the QTL-hotspot region at chromosome 4 (Fig. 7b).

Further, we analyzed the presence of DNA polymorphisms (SNPs and InDels) in the 74 transcripts located within the 100 SDW associated QTLs. The DNA polymorphisms differentiating the sets of small- and large-seeded chickpea genotypes reported in a previous study were used for this analysis[45]. At least 10 transcripts harbored DNA polymorphisms within the transcript and/or their promoter regions (Supplementary Data 9). Of these, seven transcripts harbored DNA polymorphisms in their promoter regions (Fig. 7c and Supplementary Data 9). Among these, DNA polymorphisms were identified within one or more of the cis-regulatory motifs representing the binding sites of TFs, such as AP1, PIF4, SEP3, HAT22, FLM, DREB2A, BZIP28, AtMYB12, HSFA6A, LHY1, and LFY in the promoter regions of five transcripts (Fig. 7c and Supplementary Table 6). Many of these TFs were reported to be involved in biological processes involved in cell division and elongation processes and may determine seed size/weight[45–48]. DNA polymorphisms can impact the differential affinity/binding of the TFs to their cognate cis-regulatory motifs leading to differential regulation of downstream gene expression and thus may determine the phenotypic plasticity, such as seed size/weight in chickpea. However, the identification of the exact causal gene or DNA polymorphism needs further investigation.

**QTL-associated transcripts showing abiotic stress-responsive expression.**
To identify the candidate genes that might govern abiotic stress responses, we analyzed the transcripts located within the known QTLs associated with drought and salinity stress tolerance in chickpea. We identified all the transcripts located within the drought tolerance associated known QTL regions[49–52] and analyzed their expression profiles using RNA-seq data of a drought-tolerant (Dtol; ICC 4958) and a drought-sensitive (Dsen; ICC 1882) chickpea genotype under control (CT) and drought stress (DS) conditions at early reproductive (ER) and late reproductive (LR) stages from our previous study[53]. Further, the differentially expressed transcripts were examined for the presence of DNA polymorphisms that differentiate drought-sensitive and drought-tolerant chickpea genotypes[54]. The analysis revealed a total of 65 differentially expressed transcripts that harbored the DNA polymorphisms within the transcripts and/or their promoter regions (Supplementary Data 10). The majority of them were located on the chromosomes 6 and 4 (Supplementary Fig. 19a and Supplementary Data 10). Among these, 21 differentially expressed transcripts harbored the DNA polymorphisms in their promoter regions (Fig. 8a, b). A higher number of the transcripts harboring DNA polymorphisms exhibited enhanced expression under drought stress at the LR stage in the Dsen and Dtol genotypes (Fig. 8a and Supplementary Fig. 19b). Further, the expression of many of these transcripts was found elevated to a higher level in the Dtol genotype as compared to the Dsen genotype under drought stress at ER and/or LR stages (Supplementary Data 11). Next, we identified at least 10 QTL-associated differentially expressed transcripts that harbored DNA polymorphisms within the cis-regulatory motifs, such as ABF3, SEP3, PIF4, MYB3, NYFB2, AP1, FHY1, HSFA1A, FEA4, SOC1, HB7, BZIP28, DELLA, NFYC2, KAN1, and FLM (Fig. 8b and Supplementary Table 6). Several TFs cognate to these cis-regulatory motifs have been implicated in drought stress responses/tolerance[54–57]. These transcripts and/or alleles represent important candidates for prioritization to validate their role in drought tolerance.

Next, we examined all the transcripts located within the salinity tolerance associated known QTL regions[58–60] and analyzed their expression profiles using RNA-seq data of a salinity-sensitive (Ssen; ICC V2) and a salinity-tolerant (Stol; JG 62) chickpea

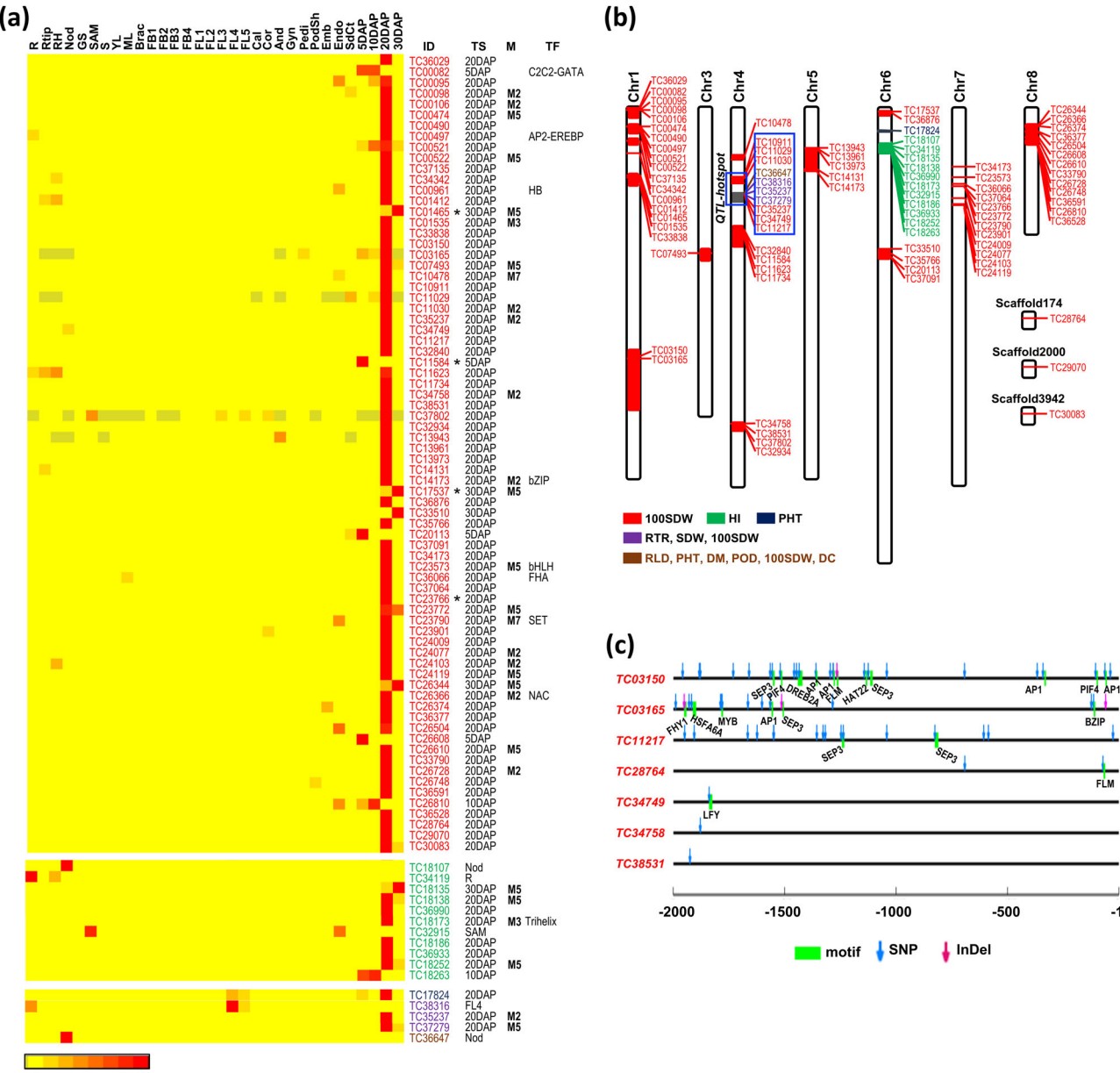

**Fig. 7 Candidate transcripts exhibiting tissue-specific expression associated with quantitative trait loci (QTLs). a** Heatmaps illustrate the tissue-specific expression of the candidate genes located within 100 seed weight (100 SDW) and other traits (HI, PHT, RTR, SDW, RLD, DM, POD, and/or DC) associated QTLs. The identifiers of the transcripts located on the QTLs of different traits are shown in different colors. The transcripts representing lncRNAs are marked with asterisks. The tissue specificity (TS) of each transcript is also indicated. Transcription factor family annotation (TF) and coexpression module (M), if any, of the transcripts are also given. The color scale at the bottom represents the expression level in Z-score. **b** Ideogram showing the genomic location of the candidate genes in the quantitative trait loci (QTLs; solid colored boxes) on the chickpea genome (vertical bars). QTLs/candidate genes for different traits are shown in different colored boxes/fonts as shown in the legend. The position of *QTL-hotspot* on chromosome 4 and the candidate genes lying within this region are marked with blue colored boxes. **c** The DNA polymorphisms (SNPs and Indels) differentiating small- and large-seeded chickpea genotypes that are located in the promoter (2 kb upstream) regions of the transcripts are shown. The *cis*-regulatory elements harboring the DNA polymorphisms are also indicated.

genotype under control (CT) and salinity stress (SS) conditions at vegetative (Veg) and late reproductive (LR) stages[53]. We identified a total of 11 differentially expressed transcripts harboring DNA polymorphisms (that differentiate salinity-sensitive and salinity-tolerant genotypes)[61] within the transcript and/or promoter regions (Supplementary Fig. 19a, c, Supplementary Table 7, Supplementary Data 9 and 11) and most of them were located on chromosome 2 (Supplementary Fig. 19a and Supplementary Table 7). All of these 11 transcripts harbored DNA polymorphisms within their transcript regions and five of

them harbored DNA polymorphisms in their promoter regions too (Fig. 8d and Supplementary Data 9). Most of these transcripts exhibited higher expression in the Stol genotype at the LR stage under salinity stress (Fig. 8c and Supplementary Fig. 19c). Further, the differential expression of many of these transcripts was higher in the Stol genotype as compared to the Ssen genotype under salinity stress (Supplementary Data 11). Among the five transcripts harboring DNA polymorphisms in their promoter regions, four harbored DNA polymorphisms in the *cis*-regulatory motifs, such as REV, SOC1, DELLA, FIE, FHY3, SVP, SEP3,

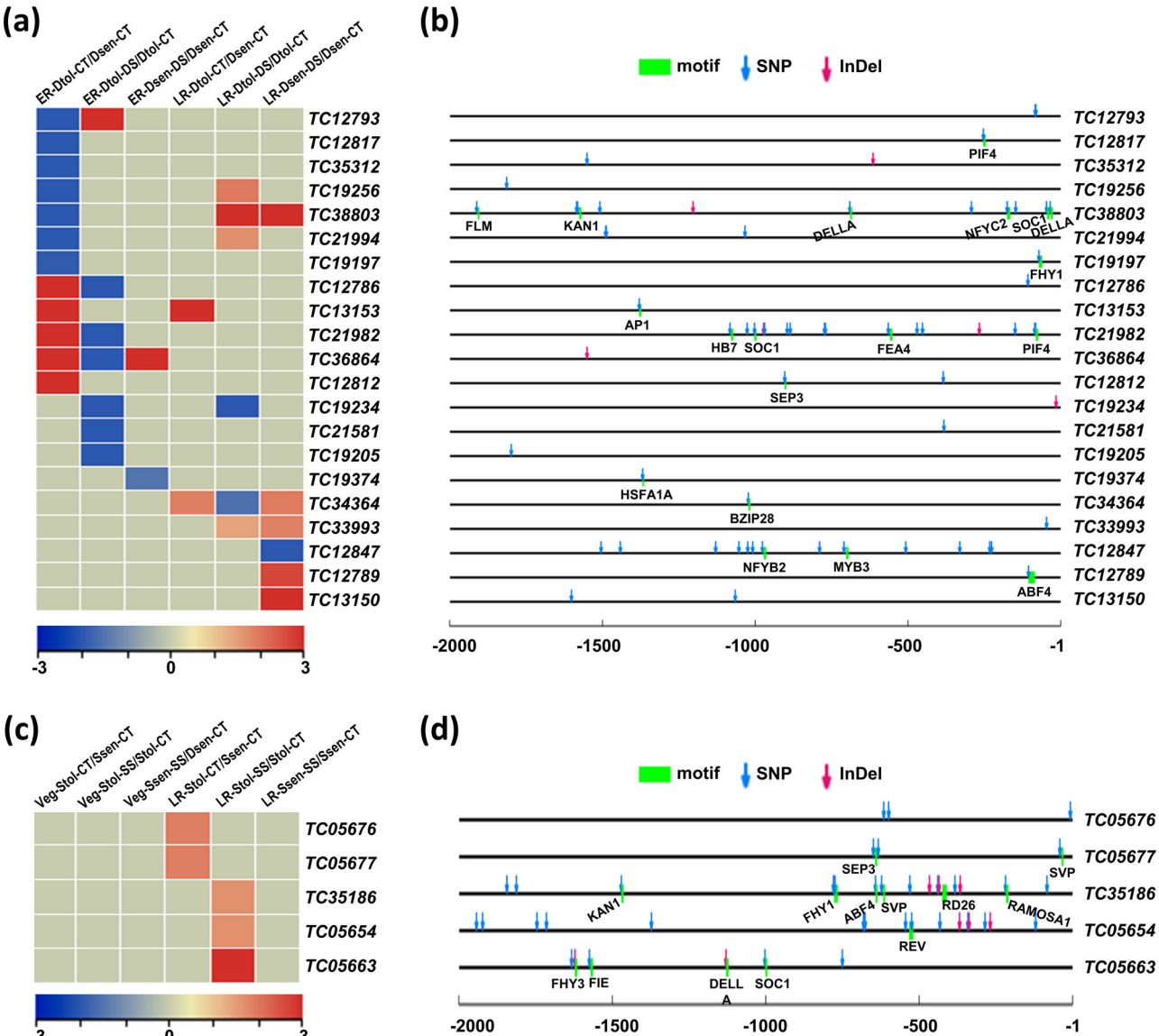

**Fig. 8 Candidate transcripts exhibiting abiotic stress-responsive expression associated with quantitative trait loci (QTLs) in the chickpea cultivars.** **a**, **c** Heatmaps showing the differential expression of transcripts located in the drought (**a**) and salinity (**c**) related QTLs and harbor DNA polymorphisms in their promoter regions. For drought stress (**a**), the differential expression within/between the drought-tolerant (Dtol) and drought-sensitive (Dsen) chickpea genotypes under drought stress (DS) or control (CT) condition at early reproductive (ER) and late reproductive (LR) stages is shown. For salinity stress (**c**), the differential expression within/between the salinity-tolerant (Stol) and salinity-sensitive (Ssen) chickpea genotypes under salinity stress (SS) or control (CT) condition at vegetative (Veg) and late reproductive (LR) stages is shown. Scales at the bottom of **a** and **c** represent log₂ fold-change differential expression. **b**, **d** The DNA polymorphisms (SNPs and InDels) differentiating the drought-tolerant and drought-sensitive (**b**), and salinity-tolerant and salinity-sensitive (**d**) chickpea genotypes that are located in the promoter (2 kb upstream) regions of the transcripts (given in **a** and **c**) are shown. The *cis*-regulatory elements harboring the DNA polymorphisms are also indicated.

RAMOSA1, RD26, ABF4, FHY and KAN1 (Fig. 8d and Supplementary Table 6). Many of these motifs represented the binding sites of TFs that have been implicated in salinity stress response/tolerance[62,63].

The QTL mapping studies have identified a *QTL-hotspot* region of ~7.74 Mb on the chromosome 4 associated with several agronomic traits, including abiotic stress responses[42]. A finer mapping further refined this region to ~2.8 Mb (Chr4: 11,020,420–13,853,257) size, which has been further split into two subregions (*QTL-hotspot_a*, Chr4: 13,239,546–13,378,761 and *QTL-hotspot_b*, Chr4: 13,393,647–13,547,009)[42]. We analyzed the differential expression of all the transcripts located within these subregions under drought and salinity stress

conditions in/between the drought/salinity sensitive and tolerant genotypes using the RNA-seq data[53]. A total of 12 transcripts located within *QTL-hotspot_a* (eight) and b (four) regions showed differential expression under drought and/or salinity stress conditions in/between the sensitive and tolerant genotypes (Supplementary Fig. 20a and Supplementary Data 11). All of these were found to be differentially expressed under drought and five of them exhibited differential expression under salinity stress conditions in the chickpea genotypes (Supplementary Fig. 20b, c). The transcripts encoding serine/threonine kinases (*TC11172, TC36647, TC11173, TC36827* and *TC36465*), protein of unknown function (*TC11179*) and protein kinase superfamily protein (*TC37396*) were found differentially expressed specifically under

drought stress conditions/genotypes (Supplementary Fig. 20b). However, the transcripts encoding DUF828 with plant pleckstrin homology-like region (*TC11187* and *TC37325*), leucine-rich repeat (*TC11193*), kinase interacting protein (*TC11196*) and homocysteine methyltransferase 2 (*TC37396*) showed differential expression under both drought and salinity stress conditions/genotypes (Supplementary Fig. 20b, c). Many of these genes have been implicated in different abiotic stress responses[64–66]. Although these transcripts represent important candidates for engineering stress tolerance in chickpea, their detailed functional validation is required.

## Discussion

The previous chickpea transcriptomes in different biological contexts based on the de novo and/or reference-based methods have been generated using only the short-read data[12,67]. The availability of long-read PacBio technology now allows the generation of full-length transcripts and isoforms[34,68]. However, due to the higher error-rate of this technology and the cost of sequencing large number of samples individually at high-depth to achieve the representation of even lowly expressed transcripts, it is advisable to use the hybrid method (combination of long-read and short-read) to generate a complete set of full-length transcripts and expression profiling. Here, we report a unified set of full-length 38,818 transcripts using >2.1 million consensus reads obtained via PacBio sequencing and ~6.5 billion Illumina reads from 32 different tissues/organs in multiple biological replicates. A significant number of novel PCGs were also identified in the reference transcriptome assembly. We also annotated a set of 5,293 high-confidence lncRNAs that represents the first comprehensive set of lncRNAs in different chickpea tissues/organs. Further, most of PCGs and lncRNAs were assigned with a putative function.

We generated the gene expression atlas of both PCGs and lncRNAs in 32 different tissues/organs representing vegetative and reproductive stages of development and were able to detect the expression of most of the transcripts due to the high-depth of sequencing (http://ccbb.jnu.ac.in/CaGEA). The expression analysis revealed the representation of a substantial proportion of even lowly expressed and tissue-specific transcripts in the reference transcriptome assembly. Overall, these observations suggest the high-quality of reference transcriptome assembly generated in this study and present a comprehensive atlas of gene expression profiles of both PCGs and lncRNAs.

RNA-seq presents a robust and sensitive tool for analyses of gene expression profiles at whole-genome level[8,9,69]. Based on the gene expression profiles of all the transcripts, we observed the clustering of tissues/organs with similar morphological features and/or representing similar/successive developmental events broadly in five groups of tissues, including root tissues, green tissues, flower development stages, flower parts and seed tissues. The identification of GO terms representing the characteristic biological processes associated with tissue-specific PCGs and lncRNAs in the five groups of tissues further highlighted their biological relevance related to tissue/organ identity during development. Although different tissues/organs revealed numerous tissue-specific expression profiles, a larger number of transcripts exhibiting tissue-specific expression patterns were detected in the 20DAP followed by nodule, endosperm and 5DAP. The higher specificity of gene expression profiles in nodules that are involved in the nitrogen fixation process has been found in a previous study too[19]. Likewise, a significantly large number of tissue-specific genes have been identified in soybean seed[36]. A substantially higher number of tissue-specific genes identified at the 20DAP stage suggest a higher transcriptional activity, which

may be attributed to the high rate of metabolism, synthesis, transport and storage of starch, proteins and oil bodies during the grain filling and seed maturation. In addition, the higher expression of transcripts involved in other biological processes, such as nitrogen fixation during seed development[70] may also be crucial for grain filling related processes. Recently, a spatiotemporal expression profiling identified several components and biological processes implicated in seed development in rice and soybean[36,71]. The significantly enriched biological process GO terms found associated with specifically expressed genes identified in different tissues reflected the common and specific biological processes underlying their development.

TFs play an important role in several growth and development related biological processes via complex gene regulatory networks[70,72,73]. We identified the members of several well-known TF families implicated in crucial developmental processes that exhibited preferential expression in different tissues/organs. The functional diversity of AP2-EREBP family TFs in several developmental processes and stress responses has been reported in chickpea and other plants[53,74,75]. The role of NF-Y TF family in the regulation of seed development-related processes is well demonstrated[76,77]. In addition, the members of other TF families, such as CCAAT, bHLH, HB, MADS-box, MYB and LOB that exhibited tissue-specificity were found implicated in several developmental processes in chickpea and other plants[12,70,72,73,78,79]. Although the exact role of most of the TFs remains unknown as of now, these results highlight the presence of unique and complex transcriptional programs regulating developmental processes in different tissues/organs in chickpea.

The coexpression and TRN analysis can provide deeper insights into the gene regulatory circuits that control specific biological processes operative in different tissues[9–13]. Our coexpression network analysis identified several modules harboring sets of transcripts with similar expression profiles that were found associated with a particular tissue or a group of related tissues in chickpea. The presence of several TF-encoding transcripts and lncRNAs in these modules indicated the complex and specific transcriptional regulatory programs associated with different tissues/organs. The TF(s) included in each module can act as the master regulator(s) that coordinate the activity of other coexpressed genes. The TRNs for the modules associated with a specific or similar group of tissue(s) reflected the TFs and target genes associated with GO terms relevant to the biological processes occurring in those tissues. Some of the components of these TRNs have already been implicated in various aspects of development relevant to a particular tissue(s) in chickpea and other plants[12,25,36,80,81]. For example, the role of MYB and WRKY TFs and plant hormones especially auxin in root and root hair development has been demonstrated[81–83]. Further, the components involved in abiotic and biotic stress responses were also identified, which is expected as the roots and root hairs are the organs that come in direct contact with soil. Likewise, many of the TFs, their binding sites and target genes identified as components of the TRN associated with 20DAP in chickpea have been reported for their role in different aspects of seed development processes and gene regulatory network in soybean too[12,36,80]. These results highlighted that construction of TRNs with the coexpressed set of genes can provide important insights into the mechanisms underlying various developmental processes and agronomic traits.

Although a few studies have suggested the role of lncRNAs in abiotic stress responses in chickpea, their role in developmental processes has not been reported so far. Here, we identified several lncRNAs as the putative components of TRNs associated with specific tissues, including root tissues, embryo and 20DAP stage of seed development. Recently, the role of lncRNAs during seed

development has been studied via their expression profiling in peanut and chickpea and their putative targets involved in transcription, cell division and plant hormone biosynthesis were identified[84,85]. In addition, the role of lncRNAs along with miRNAs in regulating key genes associated with seed and pod development have been elucidated in pigeonpea too[86]. A few lncRNAs involved in embryo development that target genes involved in hormone biosynthesis, circadian rhythm and signal transduction have also been identified in *Ginkgo biloba*[87]. These results suggest the implication of lncRNAs in governing diverse developmental processes via complex TRNs in chickpea too.

Several molecular genetic studies have identified QTLs associated with different agronomic traits in chickpea[37–44]. However, most of these studies have identified large genomic regions as QTLs that harbor several genes. The integrated analyses of molecular genetics and transcriptome datasets can provide a better method for the identification of the important candidate genes implicated in the regulation of agronomic traits. Using this integrated approach, we identified several clusters of genes associated with 100 SDW QTLs on different chickpea chromosomes and one major cluster of genes associated with HI QTLs on chromosome 6. At least 10 candidate genes were associated with 100 SDW QTLs, of which four genes were associated with QTLs of other traits (RTR, SDW, RLD, PHT, DM, POD and DC) too. These genes included TFs, transporters, protein kinases, cellulase synthase, hydrolases, gibberellin 20 oxidases, hydroxysteroid dehydrogenases, ribosomal proteins, chromatin factors, proteins of unknown function and lncRNAs. The function of many of these genes in seed development, seed size/weight determination and other aspects of plant growth and development have been reported in previous studies in chickpea and their orthologs in other plants[12,22,42,73]. The role of TFs belonging to AP2-EREBP, HB, NAC and bZIP families in seed development and seed size determination in addition to the plant growth and development in general is well known[12,73,78,88,89]. Further, the role of epigenetic modifications and chromatin modifiers in regulation of developmental processes including seed development has also been documented in chickpea and other plants[90–92]. The four lncRNAs located on the 100 SDW QTLs and expressed specifically during stages of seed development represent key candidates for functional characterization. The DNA polymorphisms within the *cis*-regulatory motifs identified in the promoter regions of tissue-specific transcripts may affect the binding/affinity of their cognate TFs[93] and thus influence the downstream gene expression and TRNs that may result in the regulation of developmental processes.

Next, we demonstrated the application of reference transcriptome assembly to identify the candidate transcripts that may govern drought and salinity stress responses. Via the integration of the known QTLs associated with drought[49–52] and salinity[58–60] tolerance, differential gene expression in/between the sensitive and tolerant genotypes under stress conditions, and DNA polymorphisms that differentiate sensitive and tolerant genotypes[54,61], at least 10 drought-responsive and four salinity-responsive transcripts were identified that harbored DNA polymorphisms within the *cis*-regulatory motifs representing the binding sites of important TFs. Many of these TFs and the target genes have been implicated in the regulation of the biological processes associated with drought and/or salinity stress responses/tolerance[54–57,62,63]. In addition, 12 drought and/or salinity stress-responsive transcripts located on the refined *QTL-hotspot*_a and b regions on chromosome 4 associated with drought tolerance were identified. The role of some of these genes in various aspects of plant growth and abiotic stress responses has been well studied in other plants[64–66]. Altogether, the candidate genes identified in our study represent important target genes involved in the regulation of

developmental programs and/or environmental cues and can be prioritized for further functional validation and identification of exact causal DNA polymorphism(s) followed by engineering agronomic traits in chickpea.

In summary, we have generated a high-quality full-length reference transcriptome assembly and expression atlas of PCGs and lncRNAs encompassing vegetative and reproductive organs/tissues in chickpea. The availability of a complete set of sequences and expression profiles of PCGs and lncRNAs throughout development presents a significant step towards chickpea functional genomics and assignment of function to both coding and non-coding components of the genome. The results highlighted the expression dynamics and tissue-specificity of PCGs and lncRNAs involved in diverse biological processes relevant to the characteristic key events in the respective tissues/organs. Further, the TRN analysis provides insights into the aspects of the gene regulation operative in different tissues/organs in chickpea. Our analyses highlighted the important contribution of lncRNAs in the developmental biology of different tissues/organs and associated lncRNAs with putative functions. Further, coexpressed lncRNA-mRNA pairs can facilitate the functional characterization of the *cis*-regulatory potential of lncRNAs. The correlation of expression profiles/modules with the QTLs revealed candidate genes that may determine important agronomic traits, including seed size/weight and abiotic stress responses, which provide a set of key target genes for further functional characterization. The datasets presented in this study will not only provide a comprehensive resource for large-scale investigations into gene function, but a gateway for genomics-enabled improvement of chickpea. The results of our study are also expected to facilitate knowledge transfer to other legumes and large-scale comparative analysis of transcriptomes in different plants.

## Methods

**Plant materials**. The plants/seedlings of chickpea (*Cicer arietinum* L.) genotype ICC 4958 were grown under culture room and field conditions as described earlier[12,25]. We collected a total of 32 tissue samples representing different organs/developmental stages in at least three independent biological replicates each. Roots (R) and shoots (S) were collected from 15-day-old seedlings grown in a culture room. The 5-day-old seedlings grown on filter papers in the Petri dish represented germinating seedlings (GS). Root tips (Rtip; 0.5-1 mm) were collected from 15-old seedlings. The root hairs (RH) were harvested from 7-day-old seedlings grown in dark as described earlier[94]. The nodules (Nod) were collected from the field-grown mature plants. The shoot apical meristems (SAM) were dissected from 21-day-old seedlings using a dissecting microscope. The flower buds and flowers were collected at four (4 mm, FB1; 6 mm, FB2; 8 mm, FB3 and 8–10 mm, FB4) and five developmental stages (FL1, flower with closed petals; FL2, mature flower with partially opened petals; FL3, mature flower with fully opened petals; FL4, mature flower with opened and faded petals; FL5, drooped flower with senescing petals), respectively, from the field-grown plants as described earlier[25]. The floral organs (calyx, corolla, androecium, gynoecium and pedicel) were collected from the partially opened flowers (FL2 stage). The leaves were collected from two developmental stages, one representing young light-green leaves (YL) near the shoot apex and the second corresponding to fully-expanded mature leaves (ML). The bracteoles (Bract) were collected from the mature plants. Seeds were sampled from four developmental stages [5, 10, 20, and 30 days after pollination (DAP)]. The pod shell and seed parts (seed coat, embryo and endosperm) were collected from the 10DAP stage pods/seeds. The tissue samples were snap-frozen in liquid nitrogen immediately after collection.

**RNA isolation**. Total RNA from all the tissue samples was extracted using TRI reagent (Sigma Life Sciences). The quality and quantity of all RNA samples were determined using agarose gel electrophoresis, Nanodrop Spectrophotometer (Thermo Fisher Scientific, Wilmington) and Bioanalyzer (Agilent Technologies, Singapore). Only high-quality RNA samples with RIN number >8.0 were used for RNA sequencing. The same RNA samples were used for Iso-seq and RNA-seq using the PacBio and Illumina platform, respectively.

**Iso-seq sequencing and high-quality transcript sequences**. RNA samples from different tissues, including S, R, YL, ML, flower bud, flower and seed stages, were pooled in equal amount to make a single pooled RNA sample for Iso-seq using PacBio platform (Pacific Biosciences, USA). Four libraries were prepared using

SMRTbell Template Prep Kit as described by the manufacturer (Pacific Biosciences). The size selection of different size cDNAs (0.8–2 kb, 2–3 kb, 3–6 kb and 5–10 kb) was done using BluePippin (Sage Science, USA). The size-selected cDNA pools were used to construct libraries according to the Iso-Seq protocol. The four libraries were sequenced in a total of six SMRT cells with a movie time of 480/600 min for each run on a PacBio RS II instrument by a commercial sequencing service provider (Agrigenome Lab Pvt. Ltd., India). The sequencing output obtained from each library was run through the Iso-Seq analysis pipeline in command line mode using pbtranscript program. Firstly, CCS (circular consensus sequence) was run using the minimum passes of 0 or greater and a minimum predicted accuracy of 0.8. Following CCS step, pbtranscript classify was run to obtain full-length non-chimeric reads (containing 5′ and 3′ adapters used in library preparation along with poly(A) tail). Further, we performed clustering of full-length reads using pbtranscript cluster for each size bin data using iterative clustering for error correction (ICE). Next, Quiver was used to obtain high-quality polished isoforms with minimum expected accuracy of 0.99. Finally, clustering of the transcripts with 99% identity over at least 95% of the alignment length was done using CD-HIT-EST v4.7 and a non-redundant set of full-length transcripts was obtained.

**RNA sequencing using Illumina platform and data pre-processing**. The RNA-sequencing libraries from different tissue samples were constructed and sequencing was done on Illumina Hi-seq platform to generate 100-nt long paired-end reads. A total of 94 libraries representing 32 tissues/organs (all tissue samples in three biological replicates except 20DAP and 30DAP in two biological replicates) were sequenced and ~31–116 million raw reads were generated for each library. The raw reads were pre-processed using NGS QC Toolkit[95] (v2.3) and 31-109 million high-quality filtered reads for each sample were obtained.

**Generation of transcriptome assembly**. High-quality reads from all the 94 samples were mapped to the reference kabuli chickpea genome (v1.0)[3] using TopHat (v2.0.0) with default parameters. The kabuli chickpea genome represents better contiguity and has been more frequently used as the reference for data analyses. A reference-based transcriptome assembly was performed using StringTie (v1.3.4d). The non-redundant set of transcripts obtained from Iso-seq was merged with the reference-based transcriptome assembly of Illumina data. The transcripts with at least 99% identity over ≥95% length of the shorter transcript were clustered using CD-HIT-EST and the longest representative in each cluster were retained in the final transcriptome. The transcripts thus generated were aligned to the kabuli chickpea genome using minimap2 using splice aware options and mapped isoforms were collapsed into transcript loci using Cupcake collapse_isoforms_by_sam.py script with a minimum of 80% coverage and 95% identity. The transcripts which did not map to the genome were further clustered and included in the final set of transcript loci referred to as final reference transcriptome assembly. The AS events represented in the complete set of transcripts were determined using the AStala-vista web tool (version 3; http://genome.crg.es/astalavista/) at default parameters.

**Functional annotation**. The annotation of final set of transcripts was performed via BLAST searches against different databases, including TAIR and UniProtKB/Swiss-Prot Viridiplantae protein databases using an e-value cut-off of ≤1e-05. GO terms for transcripts were extracted for their best hit in the UniProt database. The protein domains were identified using hmmscan (HMMER v3.2.1) against the Pfam database. The pathway annotation for the chickpea transcripts was performed using KEGG Automatic Annotation Server (KAAS, https://www.genome.jp/kegg/kaas/) against Arabidopsis database. Transcripts encoding TFs were identified using the Pfam domain profiles of 86 TF families as described earlier[24].

**LncRNA identification and functional annotation**. An initial set of putative lncRNAs were predicted using PLncPRO (v1.1)[7] with a probability cut-off of 0.8. Further, the protein-coding domain search was performed against the Pfam database using HMMER (v3.2.1) with an e-value cut-off of ≤0.001. The transcripts with no hits in the Pfam database were identified as the final set of lncRNAs. LncRNAs were classified into different types (lincRNA, natlncRNAs, exlncRNAs and intlncRNAs) based on their position with respect to PCGs using cuffcompare (v2.2.0).

To elucidate the putative function of predicted lncRNAs, their *cis*- and *trans*-target genes were predicted. The PCGs located within 10 kb flanking regions of each lncRNA were classified as *cis*-targets of the respective lncRNA(s) as reported[96]. To infer the potential *trans* targets, the Pearson correlation coefficient (PCC) was computed between lncRNAs and the PCGs based on their expression profiles. The PCGs with an absolute correlation of ≥0.97 with lncRNA(s) were identified as their *trans*-targets. The lncRNAs were assigned with putative GO terms according to their *cis* and/or *trans* PCG targets.

**LncRNA conservation analysis**. To investigate the conservation of lncRNAs among plant species, the predicted lncRNAs from different plants available in public databases were used. A total of 2,03,455 and 2,39,631 lncRNA sequences for 60 species were downloaded from GREENC v1.12 (http://greenc.sciencedesigners.com/wiki/Main_Page) and CANTATA v2.0 (http://cantata.amu.edu.pl/download.

php) databases, respectively. The predicted chickpea lncRNAs were aligned using BLASTN against the downloaded lncRNAs with an *e*-value cut-off ≤1e−05.

**Expression profiling and tissue specificity analysis**. RNA-seq reads of all 94 libraries were aligned to reference transcriptome assembly using bowtie2 (v2.3.4.1). The mapped reads were processed for quantification of normalized gene abundance in FPKM values using RSEM (v1.2.28) for each sample. Pearson correlation between the biological replicates of each sample was determined using FPKM values via corrplot utility in R package. Further, the average FPKM of each transcript in all the replicates for each of 32 different tissues/developmental stages was calculated to generate the expression atlas. The non-metric multidimensional scaling (NMDS) analysis among all the tissue samples was performed in R.

To identify tissue-specific transcripts, tissue specificity index (TSI) was calculated both for lncRNAs and PCGs as described in previous study[97]. It is based on the estimation of the *Tau* value (based on the read count in all the tissues analyzed) of each transcript, which ranged from 0 to 1, indicating the broad and specific expression, respectively. The genes with FPKM of ≥0.5 in at least one tissue were considered for the calculation of TSI. A higher TSI value indicated the specificity of transcripts towards a particular tissue. The TSI score of ≥0.9 was used to define a transcript as tissue-specific. The heatmaps for different sets of transcripts based on row-wise *Z*-score or log₂ FPKM values were generated using heatmap2 utility.

**Co-expression network analysis**. Weighted gene co-expression network analysis (WGCNA v1.61)[98] was performed as described earlier[10] to identify modules of highly correlated genes using FPKM expression measurements. The genes with average FPKM of ≥0.1 in at least one tissue sample were retained and log₂ transformed. PCC for 13,176 transcripts with variance ≥1.5 was calculated and transformed into an adjacency matrix. The soft-threshold power $\beta$ of 9 was selected with a scale-free topology index ($R^2$) of 0.86 to calculate adjacency. Topological Overlap Measure was calculated from adjacency matrix and used to cluster co-expressed genes into modules where minClusterSize was set to 30 and deepSplit to 2. Further, similar modules with a cutHeight of 0.2 were merged using the mergeCloseModules function. The eigengene value for each module was calculated using the expression data. The module-tissue relationship was established based on PCC values between each module eigengene and tissue/developmental stage. Statistical significance of the correlation was determined via calculation of *P*-value.

**Prediction of transcriptional regulatory modules**. The 2 kb upstream sequences for all the transcripts in a module were extracted and scanned for enriched sequence motifs using findMotifs.pl script in HOMER (v4.9.1). The relationship between TFs in each or a set of module(s) and the enriched motifs was predicted based on their annotations as described earlier[12]. GO terms were assigned to the transcripts and transcriptional modules (TF-cis-motif-GO term) were visualized using Cytoscape (v3.7.1).

**GO enrichment analysis**. GO enrichment analysis for the given gene sets was performed using BINGO utility in Cytoscape (v3.7.1) as described earlier[12] or AgriGO using the online portal (http://systemsbiology.cau.edu.cn/agriGOv2/). The GO terms exhibiting a corrected (after adjusting with false discovery rate) *P*-value of ≤0.05 were considered to be significantly enriched.

**Identification of transcripts located within QTLs**. The genomic coordinates of the known seed size/weight and other agronomic traits associated QTLs (100 seed weight, R–T ratio, RTR; shoot dry weight, SDW; plant height, PHT; primary branches, PBS; days to 50% flowering, DF; days to maturity, DM; pods per plant, POD; harvest index, HI and delta carbon ratio, DC) reported in previous studies[37–44] were retrieved and all the transcripts located within these QTLs were identified. The transcripts located within these QTLs and showing tissue-specific expression were shortlisted and analyzed. Likewise, we identified all the transcripts located within the known drought and salinity stress related QTLs reported in the previous studies[49–52,58–60] and analyzed further for their differential expression in different chickpea genotypes under control/stress conditions or the presence of DNA polymorphisms within the transcript and/or their promoter (2 kb upstream) regions.

**Abiotic stress-responsive expression analysis**. We analyzed the differential expression of all the transcripts located within the known drought and salinity stress related QTLs (taken from the previous studies)[49–52,58–60] in the chickpea genotypes with contrasting responses to these stresses using RNA-seq data from our previous study (Garg et al.[12]). The RNA-seq data generated from the roots of a drought-tolerant (Dtol; ICC 4958) and a drought-sensitive (Dsen; ICC 1882) chickpea genotype under control and drought stress conditions at two developmental stages [early (ER) and late reproductive (LR)] was analyzed as described[53]. Likewise, The RNA-seq data generated from the roots of a salinity-tolerant (Stol; ICC V2) and a salinity-sensitive (Ssen; JG 62) chickpea cultivars under control and salinity stress conditions at two developmental stages [vegetative (Veg) and late reproductive (LR)] was analyzed. The transcripts with a significant level of

differential expression (log$_2$ fold change of ≤−1 or ≥1 at *P* value ≤0.05) in each of the 12 comparisons, including between the two genotypes for both drought (ER-Dtol-CT/Dsen-CT and LR-Dtol-CT/Dsen-CT) and salinity (Veg-Stol-CT/Ssen-CT and LR-Stol-CT/Ssen-CT) under control conditions, and within each genotype under drought (ER-Dtol-DS/Dtol-CT, ER-Dsen-DS/Dsen-CT, LR-Dtol-DS/Dtol-CT and LR-Dsen-DS/Dsen-CT) and salinity stress (Veg-Stol-SS/Stol-CT, Veg-Ssen-SS/Ssen-CT, LR-Stol-SS/Stol-CT and LR-Ssen-SS/Ssen-CT) conditions at the two developmental stages, were identified.

**Identification of DNA polymorphisms in the QTL-associated transcripts**. We analyzed the DNA polymorphisms differentiating the large-seeded and small-seeded chickpea genotypes reported previously[45] in the tissue-specific transcripts and their promoter regions (2 kb upstream) associated with 100 SDW-related QTLs[37–44]. Likewise, we analyzed the DNA polymorphisms differentiating chickpea genotypes with contrasting responses to drought[54] and salinity[61] stresses in the stress-responsive transcripts and their promoter regions associated with drought[49–52] and salinity[58–60] stress-related QTLs, respectively. The promoters of all the transcripts harboring DNA polymorphism(s) were analyzed via PlantPAN[99] to identify the putative *cis*-regulatory elements. Further, an overlap between the position of DNA polymorphism(s) and the *cis*-regulatory element(s), if any, was determined.

**Statistics and reproducibility**. The details of the experimental design and statistical tests used in the study are described in the respective results and methods sections. The RNA-seq was performed in three independent biological replicates for 30 tissues and two biological replicates for 20DAP and 30DAP. The Pearson correlations (*r*) among the tissues/stages were calculated based on the expression level (FPKM) of the transcripts. The assessment of enrichment GO terms was performed via hypergeometric test followed by adjusting with false discovery rate implemented in BiNGO plugin available in Cytoscape. The NMDS analysis among all the tissue samples was performed in R using nonmetric mode and MASS package.

**Reporting summary**. Further information on research design is available in the Nature Research Reporting Summary linked to this article.

## Data availability

The PacBio sequencing data generated in this study are available in the SRA public repository under the bioproject accession number PRJNA613159. Illumina RNA sequencing data and the reference transcriptome assembly generated in this study have been deposited in the Gene Expression Omnibus (GEO) database under the series accession number GSE147831. Different datasets generated in this study, including reference transcriptome assembly, annotation, and expression data are available for bulk download via the webpage http://ccbb.jnu.ac.in/CaGEA. The different annotation and expression data are provided in the Supplementary Data too. The source data behind the graphs are available in Supplementary Data 2.

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

## Acknowledgements

This work was supported by the Department of Biotechnology, Government of India, New Delhi (BT/AGR/CG-PhaseII/01/2014). M.J. acknowledges the Tata Innovation Fellowship from the Department of Biotechnology, Government of India, New Delhi. The support under the Center for Computational Biology and Bioinformatics (BT/PR40251/BTIS/137/11/2021) from the Department of Biotechnology, Government of India and under the FIST scheme from the Department of Science & Technology, Government of India is also acknowledged. R.G. acknowledges infrastructural facilities from the Shiv Nadar University, Gautam Buddha Nagar and Women Excellence Award from the Science and Engineering Research Board, New Delhi. M.S.R. acknowledges Department of Biotechnology for the research associateship. We are thankful to V. Singh for help with an initial analysis of PacBio Iso-seq data, A. Dwivedi for help with generation of comparative GO enrichment maps, and N. Khemka for processing transcriptome assembly for identification of lncRNAs.

## Author contributions

M.J. and R.G. conceived and designed the study. M.J. supervised the whole study, compiled data, performed data interpretation and wrote the manuscript. R.G. contributed to data analyses and interpretation, compiling data and writing the manuscript. J.B. and M.S.R. performed most of data analyses and contributed to data compilation and manuscript writing. All authors have read and approved the manuscript.

## Competing interests

The authors declare no competing interests.
