## [Peer Review File · Communications Biology]

Reviewers' comments:

Reviewer #1 (Remarks to the Author):

Overall I am enthusiastic about this data set. It seems well conceived and comprehensive. However, I have a large number of concerns about the presentation and interpretations that I detail below.

It is important to be clear about the distinct purposes of PacBio, which is presumably to understand transcript and isoform structure and secondarily to provide a standard from which to evaluate Illumina assemblies, and Illumina, which is to generate high read count data and thus obtain expression data about a range of tissues and to facilitate interesting analyses.

First a general comment about grammar. Throughout the document there is problem with the misuse of articles: "a", "an", "the" are frequently omitted from text at locations where they are required. For example, at the top of page 9 the text should read "... a tissue-specific manner. The highest number of ...". These are two examples of many dozens of similar cases.

I recommend that the manuscript be systemically restructured and that it be reduced in length. The introduction contains an unnecessary justification of the work (see below). In the Results section, the detailed text on the status of the PCG transcriptome is too long, overly list-like and with few novel observations. I would vastly shorten these sections and compensate with summary tables. Conversely, I would retain the somewhat detailed text dealing with novel components of the data, for example (but not limited to) the lncRNA analysis.

Introduction

The entire paragraph beginning "Although quite a few studies have been performed ..." should be eliminated. The notion that previous work in chickpea may or may not be of sufficient quality is entirely the judgment of the current authors – these are decidedly NOT scientific arguments and border on personal insinuation. And the notion that all previous work in chickpea transcription has flaws is simply incorrect. In any case, one does not need to enter into this argument in the first place. I recommend focusing instead on the value of long read sequences and their novel properties relative to other methods, and the need in chickpea for a comprehensive, tissue, organ and developmental transcriptional data set. Doing so does not require impugning the work of prior authors, which is neither necessary nor polite, and it will more directly play into the strength of the data.

I do not agree that assembling full length transcript is challenging from Illumina data. Indeed it very much depends on the method by which the data was obtained, whether a high quality reference genome exists to guide assembly, and the computational capacity available to the research team. Instead, the advantage of long-read data is its contiguity, which permits a better assessment of isoform structure and frequency.

The choice of manuscripts that the authors cite (10,20, 27-30) related to chickpea transcriptional data is rather narrow and I think avoids the most significant and in some cases recent publications. Please consult the literature and update as appropriate.

Results

From the beginning of the Results section through first paragraph on page 9, most of the text is simple accounting: i.e., numbers associated with various factors (number of libraries, total bp, redundant bp, N50, type of sequencing, numbers of reads, etc.,). There is no biological question being asked, which is inherent to this type of study. As a consequence, from the perspective of this reviewer, such details are not worthy inclusion in the text and instead should be relegated to a simple summary figure(s) or table(s) included in the supplementary materials. Thus I strongly recommend a

deep revision of the first portion of the manuscript, moving much of the information to a table and shortening the text to highlight the important summary points. I note that while Figure 4 provides a nice and rather elegant summary of the PCG data, it is almost an afterthought in this section, whereas I think it should be emphasized. Reprioritize the data and arguments and make this section more concise, please.

Similarly, I wonder if the section on GO terms (beginning on page 9) and their distributions might be better summarized in a table. The problem for this reader is that the 6 paragraphs, spanning 2 full pages, essentially represent a list of statements about GO terms, with limited insight gained. Moreover, the novelty of this section is low because similar collections of data have been described in uncountable numbers of previous manuscripts (i.e., nothing new here in these regards). In summary, a well-conceived table accompanied by concise and thoughtful text could be significantly more palatable to the reader and impactful, I think.

Conversely, the section on lncRNAs is very interesting (from mid-page 11). Given the high species-specificity of lncRNAs, these data given unique insight into chickpeas potential regulatory schema. This is one of the most novel parts of the manuscript, and I wonder if some of the related supplemental figures should instead be in the main text, for example Fig S13?

At several points in the manuscript the authors equate their analyses as "proof" of function. Thus I disagree with emphasis in the sentence, "lncRNAs were also found to be involved in several important biological processes. ...". None of the analyses in this manuscript (or any study of transcription) provide proof of function. Rather, while transcriptional data may suggest that certain functions exist, they fall short of molecular proof. I would modify the above sentence and instead write "lncRNAs were implicated in several important biological processes." There are many places in the manuscript where with a similar tone that need to be changed, that correlations and patterns are not conflated with proof.

I question whether the paragraph on page 13 describing GO enrichment in gene expression modules is necessary. All of these relationships can be expressed in a table, and summarized with rather limited text. Writing lists of relationships has the very real effect of causing the reader to be disinterested. The conclusions are not novel and the list does not give insight. I do not dispute that these relationships are important, but a table with brief text is a better format.

I very much like the section on TRNs and lncRNAs, starting on page 13, because it begins to integrate the various analyses – PCGs, lncRNAs, cis-element predictions, potential functional pathways, etc. However, the text is rather long and could more concise.

QTL analysis paragraphs

I like the section relating gene expression to QTL data, but one needs to avoid equating linkage with function. This type of "star gazing" can be misleading – while potentially dozens of interesting genes may reside in a particular genetic interval, only one will be causal. Thus one can devise numerous plausible scenarios, despite the fact that the majority are wrong – in such instances, science quickly gives way to storytelling. Similarly, because there are many genes in equal linkage to the trait, and only one if correct, the GO enrichment analyses are non-sensical (have no value) and should be removed. The alternative, that the associated phenotype derives from the combined activities of genes in a haplotype block, is highly improbable. Throughout this section the arguments should be tempered and the assumptions and limitations should be made clear.

Moreover, in each of the considered QTL intervals, please specify the physical size of the considered window, which should be the full extent of the genetically defined region, not an arbitrary region. Indicate the total number of genes in that interval. Keep in mind that one has no idea of where the causal gene lies – there is no "center" of the QTL, only the flanking recombinant boundaries, and

within the non-recombinant region one can gain neither orientation nor proximity.

Discussion

The entire first and second paragraphs of the Discussion are unnecessary. Comparisons to earlier work that are made only to highlight the importance of the current work create a vacuous argument and should be avoided. Moreover, while I do not dispute the importance of the long read data, the vast majority of the manuscript focuses solely on the Illumina data. In this regard, it would have been preferable if Results made clear the specific and important contribution of the long read data. This is currently vague at best.

I suggest that the first paragraph of the Discussion should highlight the major contributions of the current manuscript. There is certainly merit in emphasizing the breadth of the current analysis, the insights afforded by combining PacBio and Illumina, etc. But focus on the unique insights gained, not on whether, in the authors' opinion, the previous work in this area (which is only partially cited) is meritorious or complete.

I would begin the Discussion with the content of the third paragraph, starting on line 617.

The section of the Discussion on candidate genes in QTL regions has the same concerns that I expressed above regarding the corresponding region of the Results section. Many compelling stories are unhelpful when only one can be true. F-box genes for example can have dramatic phenotypic effects, but the sheer number of F-box proteins means they will be present in a very large number of genetic intervals simply by chance. False positive rate is high. To the uninformed reader, or one who does not think critically, the effect can be hypnotic and the quality of the science and its communication is degraded in proportion. Please stay faithful to what the data says, avoid unwarranted speculation, and when speculating be clear about the assumptions and limitations.

Reviewer #2 (Remarks to the Author):

The manuscript entitled « An integrated transcriptome atlas mapping the regulatory network of coding and long non-coding RNAs provides important genomics resource in chickpea” by Jain et al. reports on the development of a full-length gene expression atlas in chickpea. Both long and short read RNA-sequencing of a collection of different plant tissues have been assembled, and protein coding genes as well as long non coding RNA expression assessed. Co-expression networks have been defined and candidate genes below major chickpea QTL have been pointed out.

The manuscript present interesting original data but would deserve more in-depth analysis of two important aspects:

- Iso-seq allows the identification of gene families' isoforms as well as alternative transcripts resulting from alternative splicing; I would have appreciated a more in-depth analysis of these isoforms: what was the progress made in identifying these isoforms thanks to this method; which molecular functions presented more isoforms than others, what about the differential expression of alternative transcripts;
- The tentative identification of relevant candidate genes below QTL is interesting but more information is needed to describe these regions; as well as more information about potential causal polymorphisms in the candidate genes: what are the sequences of the parental lines for the candidate genes? What are the expected impacted of mutations?

The manuscript is generally well-written, however, the form of the manuscript could be improved:

- Remove most “Further” from the text
- Check the superlatives: THE largest ..., THE highest ...
- Several “However” can be deleted (line 240, 244, ...)

Specific comments:

Line40: grain legumes

Line 41-42: give references

Line44: what is the meaning of blueprint here?

Line 46: replace understanding by describing, assessing, ...

Line 50; meaning of "broad range of expression"?

Line 67: including the use of (i) transcriptome..., (II) only one ...

Line 67: remove generated

Line 70: remove included

Line 79: is an excellent ...

Line 96: paired end?

Line 98-99: indicate depth of sequencing

Line 109-110: please correct the position of comma: number not clear

Line 111: there are two chickpea reference genome (kabuli and desi); why use only the one from Varshney et al.?

Line 121: what about the 3000 missing PCG as compared with the genome? Are these artifacts from the genome? If not, why have the authors not included these in the reference transcriptome?

Line 133: what is AU?

Line 193: delete "of the total"

Line 202: not clear why and how were calculated these correlations? Why collections and not ANOVA?

Line 215: were expressed at very low to low ...

Line 216: not clear what means 50% of total transcripts? 50 of total number? 50% of total expression? How were the expression of the different libraries normalized?

Line 223: based on Pearson's correlations. Most ...

Line 240: not clear how was the TSI calculated

Line 254-Line 498: these paragraphs are too long and sometimes redundant; this information should be condensed and a table could be prepared to summarize most of this info (for example, the info from line 400 to 498).

Line 261: GO terms related to immune ...

Line 265: nothing in green tissues? If not, worth mentioning it.

Line 326: how did you find the GO terms for lncRNAs? Or is it GO terms of putative targets? Specify.

Line 367-370: delete.

Line 402: give the meaning of TRN here

Line 550: present the data that allowed to reduce the confidence interval to 2.8 Mb;

Line 552: not clear why this RNAseq data was not used to improve the gene atlas;

Line 558: what are the 12 comparisons?

Line 568: at whole genome level

Line 571: what means majorly found implicated? Reword.

Line 578-615: too descriptive; shorten and highlight the major breakthrough of your results;

Line 698: No, these results do not indicate that lncRNAs play important role in regulation; they just show that lncRNA expression are correlated with other genes ...

Line 708-709: what do you mean here? Why should the number of candidate depend on the number of studies? It should rather depend on the size of the confidence interval.

Line 716-721: provide other evidence; for example, provide the sequence of parental alleles for these genes, and their putative impact on the phenotype;

Line 780: why 3-6 replicates mentioned here and only three in later places? How were the three chosen from the 6?

Figure 4: quid GT? No circle for GT?

Reviewer #3 (Remarks to the Author):

Summarize the main findings of the study:

The study generated a full-length chickpea ref. transcriptome and expression resource of coding and noncoding transcripts. It contains an impressive dataset of expression profile of 32 tissues. They found a range of tissue-specific and dynamic PCGs and lncRNA involved in different plant development process. TRN coexpression network analysis also provide insight in gene regulation. They also pinpointed a few candidate genes expressed in seed from previous QTL mapping studies.

Major concerns:

1, Data availability/accessibility

As the MS was presented as a resource paper, readers may want to access the expression level of transcripts easily. Although, the authors have included the FPKM matrix in the supplementary, I strongly recommend the authors upload those information into a JBrowse on a public database such as <https://legumeinfo.org> so that readers can easily find out the expression level of the genes they are interesting in.

2, The expression profile of the genes/transcripts in the QTL-hotspot is based on the ~2.8Mb region. A more recent paper has further narrowed down this to 113kb. The authors shall revise their analyses based on that. "Fine mapping studies identified a 113 kb region within "QTL-hotspot_a" for seed weight and drought related traits in chickpea"

3, The majority of the genes are highly expressed (several hundred-fold in some cases) in 20DAP tissue compared to other tissues and even 10 DAP, 30 DAP tissue closely related to 20 DAP. And almost one third of the tissue specific transcripts belong to 20DAP (2472/7903). It is important to discuss this further. Some genes without obvious function in floral, seed development such as nitrogen fixation (Ca_TC26389) were highly expressed in 20DAP. What could be the reason?

Minor problems:

-L100 One advantage of the Pacbio/IsoSeq is its ability to detect alternative splicing events, but it seems that the MS hasn't describe the Analysis of Alternative Splicing properly. Such as location and distribution of Alternative Splicing, GO term of genes having AS..

L206-207 What is the cut-off of FPKM was used to declare "expressed"

L413-415 The tissues in this study were not sampled under stress. I am surprised that a lot of stress responsive PCGs and lncRNAs were found in the root tissue.

L489 New paragraph starting from "The M6 and M7.."

L502 Why do the transcripts need to be tissue-specific expression? I assume there will be hormone signalling/responsive genes controlling 100 SDW, but they don't need to be specifically expressed in seed.

L505 TableS7 need references for the QTL

-L807 Tissues from the same plant were used for Pacbio and illumina?

-L866 The authors may need to justify using 10kb flanking region instead of other lengths

-Fig9 Comparison of ER-Dtol-CT/Dsen-CT doesn't seem to have anything to do with drought stress.

Can the authors provide Dtol-DS/Dsen-DS for ER, LR and Veg? Those are the drought responsive one

- In supplemental Data Set 1 &2, there is only Transcript id, however, it is also very important to show the chickpea gene id or position from the CDC Frontier v1.0 gene models as readers are likely want to know the expression level of chickpea genes they are interested in. There are many transcriptome assemblies available in chickpea, few are available for visualisation in JBrowse due to lack of information or other reasons. It is really a waste of resource for the Chickpea community.

Editor Comments:

Your manuscript entitled "An integrated transcriptome atlas mapping the regulatory network of coding and long non-coding RNAs provides important genomics resource in chickpea" has now been seen by 3 referees, whose comments are appended below. You will see from their comments copied below that while they find your work of potential interest, they have raised quite substantial concerns that must be addressed. In light of these comments, we cannot accept the manuscript for publication, but would be interested in considering a revised version that addresses these serious concerns.

We hope you will find the referees' comments useful as you decide how to proceed. Should further experimental data or analysis allow you to address these criticisms, we would be happy to look at a substantially revised manuscript. However, please bear in mind that we will be reluctant to approach the referees again in the absence of major revisions.

Reply: Many thanks for your encouraging comments. The comments and suggestions by the Reviewers are indeed helpful in improving the MS. We have made sincere efforts to revise the manuscript taking into consideration all the comments.

In particular, you would need to pay special attention to the following points for us to contact our referees again:

1. As recommended by Reviewer #1 please provide context on the use of PacBio data. In addition, expand the isoform analysis as suggested by Reviewers #2 and #3.

Reply: We have now provided the context on use of PacBio data and expanded the isoform analysis as suggested by the Reviewers.

2. Reviewer #1 provides some suggestions regarding restructuring of the manuscript that should be considered. In particular, ensuring the biological questions are clearly evident and data lists are reduced to highlight just the salient points in the text (see specific comments on sequencing statistics and GO terms).

Reply: We have now restructured the manuscript and reduced the text taking into consideration all the comments of the Reviewers (specifically Reviewer #1).

3. Please consider the suggestion by Reviewer #1 to provide a greater focus on the lncRNA analysis and to expand and provide more detail on the QTL analysis as suggested by Reviewers #1 and #2.

Reply: We have now provided greater focus on the lncRNA analysis and provided more details of the QTL analysis in the revised manuscript as suggested by the Reviewers.

4. Please follow the suggestions by Reviewer #1 on restructuring the manuscript, condensing the text in individual results sections, reducing the discussion and removing negative comments on previous publications in the introduction. Please also expand the literature cited.

Reply: We have now revised all the said sections, including condensing the text in individual results sections, reducing the discussion, removing negative comments on previous publications in the introduction and expansion of the literature cited.

5. As suggested by Reviewer #3, please consider adding the expression data to a public repository for ease of access. This will ensure transparency and reproducibility of the analysis, whilst enabling other researchers to further interrogate these valuable datasets.

Reply: We have now provided access to the whole data via a webpage on Chickpea Expression Atlas (<https://cbb.jnu.ac.in/CaGEA/>) for ease of access and enabling other researchers to further interrogate these datasets.

6. Please expand the discussion regarding the number of genes shown to be differentially expressed at 20 DAP as suggested by Reviewer #3.

Reply: We have now expanded the discussion on the genes differentially expressed at 20 DAP as suggested by Reviewer #3.

7. All reviewers raise concerns regarding the language and grammar. In particular, the authors should be careful with terminology to ensure they aren't overstating their findings as commented by Reviewer #1 regarding "proof" of function. Please revise accordingly.

Reply: Thanks for the comment. We have now revised the manuscript thoroughly to correct the language/grammar and avoided overstatement of the findings.

8. Please do also carefully consider all the minor points highlighted by the reviewers.

Reply: We have revised the manuscript taking into consideration all the minor points highlighted by the Reviewers.

Reviewers' comments:

Reviewer #1 (Remarks to the Author):

Overall I am enthusiastic about this data set. It seems well conceived and comprehensive. However, I have a large number of concerns about the presentation and interpretations that I detail below.

Reply: Thanks for the appreciation. We have now addressed all the concerns about the presentation and interpretations throughout the manuscript.

It is important to be clear about the distinct purposes of PacBio, which is presumably to understand transcript and isoform structure and secondarily to provide a standard from which to evaluate Illumina assemblies, and Illumina, which is to generate high read count data and thus obtain expression data about a range of tissues and to facilitate interesting analyses.

Reply: We have now clarified the distinct purposes of using PacBio and Illumina datasets in the Introduction (page 4; lines 75-77) and Discussion (page 17; lines 525-530) sections.

First a general comment about grammar. Throughout the document there is problem with the misuse of articles: "a", "an", "the" are frequently omitted from text at locations where they are

required. For example, at the top of page 9 the text should read "... a tissue-specific manner. The highest number of ...". These are two examples of many dozens of similar cases.

Reply: We have made a sincere effort to make the corrections in the grammar (including the correct usage of articles at relevant places) throughout the manuscript, as suggested.

I recommend that the manuscript be systemically restructured and that is be reduced in length. The introduction contains an unnecessary justification of the work (see below). In the Results section, the detailed text on the status of the PCG transcriptome is too long, overly list-like and with few novel observations. I would vastly shorten these sections and compensate with summary tables. Conversely, I would retain the somewhat detailed text dealing with novel components of the data, for example (but not limited to) the lncRNA analysis.

Reply: We have now restructured the manuscript systematically and removed/shortened some of the sections keeping major emphasis on the novel components of the data. For example, we have shortened the text in the sections "Generation of full-length reference transcriptome assembly", "Functional annotation of PCGs and lncRNAs", "Expression dynamics across different tissues/organs in chickpea", "Tissue specificity of PCGs and lncRNAs, and their functional relevance" and "Co-expression network and co-expressed modules" (pages 4-10) and retained the text dealing with novel components of the data, for example, lncRNA analysis, transcriptional regulatory networks analysis, and QTL-associated transcripts showing tissue-specific and abiotic stress-responsive expression (pages 11-16).

Introduction

The entire paragraph beginning "Although quite a few studies have been performed ..." should be eliminated. The notion that previous work in chickpea may or may not be of sufficient quality is entirely the judgment of the current authors – these are decidedly NOT scientific arguments and border on personal insinuation. And the notion that all previous work in chickpea transcription has flaws is simply incorrect. In any case, one does not need to enter into this argument in the first place. I recommend focusing instead on the value of long read sequences and their novel properties relative to other methods, and the need in chickpea for a comprehensive, tissue, organ and developmental transcriptional data set. Doing so does not require impugning the work of prior authors, which is neither necessary nor polite, and it will more directly play into the strength of the data.

Reply: Many thanks for the comment. We agree with the Reviewer's suggestions and have now removed/moderated the said sentences from the Introduction section and focused only on highlighting the strength of the current study (pages 3-4; lines 72-80).

I do not agree that assembling full length transcript is challenging from Illumina data. Indeed it very much depends on the method by which the data was obtained, whether a high quality reference genome exists to guide assembly, and the computational capacity available to the research team. Instead, the advantage of long-read data is its contiguity, which permits a better assessment of isoform structure and frequency.

Reply: Thanks for the suggestion. We have now moderated the said statements and added on the advantage of long-read data (page 4; lines 75-77).

The choice of manuscripts that the authors cite (10,20, 27-30) related to chickpea transcriptional data is rather narrow and I think avoids the most significant and in some cases recent publications. Please consult the literature and update as appropriate.

Reply: Many thanks for the comment. The references in this section have been updated with citation of other relevant publications too (page 3; line 60).

Results

From the beginning of the Results section through first paragraph on page 9, most of the text is simple accounting: i.e., numbers associated with various factors (number of libraries, total bp, redundant bp, N50, type of sequencing, numbers of reads, etc.). There is no biological question being asked, which is inherent to this type of study. As a consequence, from the perspective of this reviewer, such details are not worthy inclusion in the text and instead should be relegated to a simple summary figure(s) or table(s) included in the supplementary materials. Thus, I strongly recommend a deep revision of the first portion of the manuscript, moving much of the information to a table and shortening the text to highlight the important summary points. I note that while Figure 4 provides a nice and rather elegant summary of the PCG data, it is almost an afterthought in this section, whereas I think it should be emphasized. Reprioritize the data and arguments and make this section more concise, please.

Reply: Many thanks for the suggestion. We have now shortened the descriptions/results in the initial results sections to retain the minimal details and at the same time keep in view the non-specialist readership. A detailed summary of the data and different statistics are given in Supplementary Tables S1-S4. Moreover, we have now emphasized on the sections describing the tissue-specific expression of PCGs and lncRNAs, and their functional relevance, as suggested.

Similarly, I wonder if the section on GO terms (beginning on page 9) and their distributions might be better summarized in a table. The problem for this reader is that the 6 paragraphs, spanning 2 full pages, essentially represent a list of statements about GO terms, with limited insight gained. Moreover, the novelty of this section is low because similar collections of data have been described in uncountable numbers of previous manuscripts (i.e., nothing new here in these regards). In summary, a well-conceived table accompanied by concise and thoughtful text could be significantly more palatable to the reader and impactful, I think.

Reply: Many thanks for the comment. A sincere effort has been made to concise the text in the above-mentioned section (pages 8-9; lines 233-266) to highlight only the notable observations. The list of enriched GO terms is provided in the Supplementary Figures S14 and S16 via heatmaps.

Conversely, the section on lncRNAs is very interesting (from mid-page 11). Given the high species-specificity of lncRNAs, these data given unique insight into chickpeas potential regulatory schema. This is one of the most novel parts of the manuscript, and I wonder if some of the related supplemental figures should instead be in the main text, for example Fig S13?

Reply: Many thanks for the comment. We have now shifted the supplemental Figure S13 in the main text (Fig. 3d).

At several points in the manuscript the authors equate their analyses as "proof" of function. Thus, I disagree with emphasis in the sentence, "lncRNAs were also found to be involved in several important biological processes. ...". None of the analyses in this manuscript (or any study of transcription) provide proof of function. Rather, while transcriptional data may suggest that certain functions exist, they fall short of molecular proof. I would modify the above sentence and instead write "lncRNAs were implicated in several important biological processes." There are many places in the manuscript where with a similar tone that need to be changed, that correlations and patterns are not conflated with proof.

Reply: Thanks for the suggestion. We have now modified the said sentence and moderated the similar statements at other places too, as suggested.

I question whether the paragraph on page 13 describing GO enrichment in gene expression modules is necessary. All of these relationships can be expressed in a table, and summarized with rather limited text. Writing lists of relationships has the very real effect of causing the reader to be disinterested. The conclusions are not novel and the list does not give insight. I do not dispute that these relationships are important, but a table with brief text is a better format.

Reply: We agree with the suggestion and found the description of GO enrichment at this place irrelevant, as it is already described in the next section on TRNs (page 11). Therefore, we have now removed the said paragraph from the section "Co-expression network and co-expressed modules" (page 10-11).

I very much like the section on TRNs and lncRNAs, starting on page 13, because it begins to integrate the various analyses – PCGs, lncRNAs, cis-element predictions, potential functional pathways, etc. However, the text is rather long and could more concise.

Reply: Many thanks for the appreciation. We have now shortened the text and made it more concise.

QTL analysis paragraphs

I like the section relating gene expression to QTL data, but one needs to avoid equating linkage with function. This type of "star gazing" can be misleading – while potentially dozens of interesting genes may reside in a particular genetic interval, only one will be causal. Thus one can devise numerous plausible scenarios, despite the fact that the majority are wrong – in such instances, science quickly gives way to storytelling. Similarly, because there are many genes in equal linkage to the trait, and only one if correct, the GO enrichment analyses are non-sensical (have no value) and should be removed. The alternative, that the associated phenotype derives from the combined activities of genes in a haplotype block, is highly improbable. Throughout this section the arguments should be tempered and the assumptions and limitations should be made clear.

Reply: Many thanks for the comment. We have now extended our QTL based analysis (both tissue-specific and abiotic stress-responsive expression) as per the suggestions of the other Reviewers and also moderated the sentences for more clarity and to avoid overinterpretations. Further, we have now removed the GO enrichment analyses from these sections, as suggested.

Moreover, in each of the considered QTL intervals, please specify the physical size of the considered window, which should be the full extent of the genetically defined region, not an arbitrary region. Indicate the total number of genes in that interval. Keep in mind that one has no idea of where the causal gene lies – there is no "center" of the QTL, only the flanking recombinant boundaries, and within the non-recombinant region one can gain neither orientation nor proximity.

Reply: We have now provided the details of the QTLs and the transcripts located within these intervals in the revised study in the Supplementary Tables (S7-S10).

Discussion

The entire first and second paragraphs of the Discussion are unnecessary. Comparisons to earlier work that are made only to highlight the importance of the current work create a vacuous argument and should be avoided. Moreover, while I do not dispute the importance of the long read data, the vast majority of the manuscript focuses solely on the Illumina data. In this regard, it would have been preferable if Results made clear the specific and important contribution of the long read data. This is currently vague at best.

Reply: Many thanks for the suggestion. We have now removed the first and second paragraphs in the revised manuscript.

I suggest that the first paragraph of the Discussion should highlight the major contributions of the current manuscript. There is certainly merit in emphasizing the breadth of the current analysis, the insights afforded by combining PacBio and Illumina, etc. But focus on the unique insights gained, not on whether, in the authors' opinion, the previous work in this area (which is only partially cited) is meritorious or complete.

Reply: We have now focused on highlighting the importance of the current work in the first paragraph of the Discussion section (page 17, lines 523-546), as suggested.

I would begin the Discussion with the content of the third paragraph, starting on line 617.

Reply: Needful has been done.

The section of the Discussion on candidate genes in QTL regions has the same concerns that I expressed above regarding the corresponding region of the Results section. Many compelling stories are unhelpful when only one can be true. F-box genes for example can have dramatic phenotypic effects, but the sheer number of F-box proteins means they will be present in a very large number of genetic intervals simply by chance. False positive rate is high. To the uninformed reader, or one who does not think critically, the effect can be hypnotic and the quality of the science and its communication is degraded in proportion. Please stay faithful to what the data says, avoid unwarranted speculation, and when speculating be clear about the assumptions and limitations.

Reply: We have now moderated the sentences describing the candidate genes identified based on the revised QTL based analysis (page 20-21; lines 637-666). We have tried our best to avoid unwarranted speculations.

Reviewer #2 (Remarks to the Author):

The manuscript entitled « An integrated transcriptome atlas mapping the regulatory network of coding and long non-coding RNAs provides important genomics resource in chickpea” by Jain et al. reports on the development of a full-length gene expression atlas in chickpea. Both long and short read RNA-sequencing of a collection of different plant tissues have been assembled, and protein coding genes as well as long non coding RNA expression assessed. Co-expression networks have been defined and candidate genes below major chickpea QTL have been pointed out.

Reply: Many thanks for a comprehensive review of the MS and encouraging comments.

The manuscript present interesting original data but would deserve more in-depth analysis of two important aspects:

-Iso-seq allows the identification of gene families’ isoforms as well as alternative transcripts resulting from alternative splicing; I would have appreciated a more in-depth analysis of these isoforms: what was the progress made in identifying these isoforms thanks to this method; which molecular functions presented more isoforms than others, what about the differential expression of alternative transcripts;

Reply: Many thanks for the comment. We have now incorporated the isoform level analysis including different types of isoforms represented in the transcriptome and their associated functional GO terms (lines 102-113; Figure S2).

-The tentative identification of relevant candidate genes below QTL is interesting but more information is needed to describe these regions; as well as more information about potential causal polymorphisms in the candidate genes: what are the sequences of the parental lines for the candidate genes? What are the expected impacted of mutations?

Reply: Many thanks for the comment. We have now added more information about the QTL based analysis in the text and Supplementary Tables (S7-S10). In addition, we have included the integrated analysis of DNA polymorphisms related to seed size/weight (lines 438-452) and abiotic (drought and salinity) stress traits (lines 463-479 and 484-498) from the previous studies with gene expression datasets. The results provide insights into the influence of DNA polymorphisms on differential gene expression and thus the phenotype.

The manuscript is generally well-written, however, the form of the manuscript could be improved:

- Remove most “Further” from the text
- Check the superlatives: THE largest ..., THE highest ...
- Several “However” can be deleted (line 240, 244, ...)

Reply: Thanks for the comment. We have made a sincere effort to improve the writing of the manuscript and made all the corrections.

Specific comments:

Line40: grain legumes

Line 41-42: give references

Line44: what is the meaning of blueprint here?
Line 46: replace understanding by describing, assessing, ...
Line 50; meaning of “broad range of expression”?
Line 67: including the use of (i) transcriptome..., (II) only one ...
Line 67: remove generated
Line 70: remove included
Line 79: is an excellent ...
Line 96: paired end?
Line 98-99: indicate depth of sequencing
Line 109-110: please correct the position of comma: number not clear

Reply: The needful has been done to address all the above comments.

Line 111: there are two chickpea reference genome (kabuli and desi); why use only the one from Varshney et al.?

Reply: The kabuli genome (Varshney et al., 2013) is more contiguous and frequently used in many other studies. Thus, we have also used the kabuli genome in our study as a reference so that the output can easily be integrated with other studies by the scientific community. Comment on this has been added in the manuscript now (lines 769-770).

Line 121: what about the 3000 missing PCG as compared with the genome? Are these artifacts from the genome? If not, why have the authors not included these in the reference transcriptome?

Reply: Many thanks for the comment. The missing PCGs may represent the annotation artefacts and/or are not represented in our transcriptome assembly. Comment on this has been added in the manuscript text now (lines126-128).

Line 133: what is AU?
Line 193: delete “of the total”

Reply: Needful has been done to address/correct the above comments.

Line 202: not clear why and how were calculated these correlations? Why collections and not ANOVA?

Reply: The reproducibility of the expression data among biological replicates was estimated via Pearson correlation. It has been clarified now.

Line 215: were expressed at very low to low ...
Line 216: not clear what means 50% of total transcripts? 50 of total number? 50% of total expression? How were the expression of the different libraries normalized?
Line 223: based on Pearson’s correlations. Most ...

Reply: Needful has been done to address the above comments for better clarity.

Line 240: not clear how was the TSI calculated

Reply: The relevant details are given in lines 824-830.

Line 254-Line 498: these paragraphs are too long and sometimes redundant; this information should be condensed and a table could be prepared to summarize most of this info (for example, the info from line 400 to 498).

Reply: Thanks for the comment. The text in the mentioned paragraphs has been condensed to highlight only the important observations.

Line 261: GO terms related to immune ...

Line 265: nothing in green tissues? If not, worth mentioning it.

Line 326: how did you find the GO terms for lncRNAs? Or is it GO terms of putative targets? Specify.

Line 367-370: delete.

Line 402: give the meaning of TRN here

Reply: Needful has been done to address all the above comments. The corrections have been made and additional information has been provided at the relevant places.

Line 550: present the data that allowed to reduce the confidence interval to 2.8 Mb;

Reply: We have now added the relevant lines (499-503).

Line 552: not clear why this RNAseq data was not used to improve the gene atlas;

Reply: The said RNA-seq data was generated from different chickpea genotypes/cultivars with contrasting responses to drought/salinity stresses. However, the current study is focused on the transcriptome analysis in the ICC4958 chickpea genotype during development using both Iso-seq and RNA-seq data generated from the same cultivar. Further, a high-sequencing depth has already been achieved in the current study to obtain a high-quality transcriptome assembly.

Line 558: what are the 12 comparisons?

Reply: The relevant details given in lines 884-890.

Line 568: at whole genome level

Line 571: what means majorly found implicated? Rerword.

Reply: The corrections have been made.

Line 578-615: too descriptive; shorten and highlight the major breakthrough of your results;

Reply: Thanks for the comment. We have now shortened the text between the said lines to highlight only the important observations.

Line 698: No, these results do not indicate that lncRNAs play important role in regulation; they just show that lncRNA expression are correlated with other genes ...

Reply: We have now moderated the sentence, as suggested.

Line 708-709: what do you mean here? Why should the number of candidate depend on the number of studies? It should rather depend on the size of the confidence interval.

Reply: Thanks for the comment. We have now made the correction.

Line 716-721: provide other evidence; for example, provide the sequence of parental alleles for these genes, and their putative impact on the phenotype;

Reply: We have now integrated the DNA polymorphisms data from previous studies with the gene expression data reported in this study in the developmental context and gene expression data from the previous study for the abiotic stress responses, and showed their plausible influence on the gene expression and thus the phenotype. However, these observations require further experiments for validations, which is beyond the scope of this manuscript.

Line 780: why 3-6 replicates mentioned here and only three in later places? How were the three chosen from the 6?

Reply: We had harvested the samples in 3-6 biological replicates for different tissues and used three replicates for each of them for RNA-seq experiment. We have now revised the statement.

Figure 4: quid GT? No circle for GT?

Reply: Correction has been done.

Reviewer #3 (Remarks to the Author):

Summarize the main findings of the study:

The study generated a full-length chickpea ref. transcriptome and expression resource of coding and noncoding transcripts. It contains an impressive dataset of expression profile of 32 tissues. They found a range of tissue-specific and dynamic PCGs and lncRNA involved in different plant development process. TRN coexpression network analysis also provide insight in gene regulation. They also pinpointed a few candidate genes expressed in seed from previous QTL mapping studies.

Reply: Many thanks for a thorough and critical review of the MS.

Major concerns:

1, Data availability/accessibility

As the MS was presented as a resource paper, readers may want to access the expression level of transcripts easily. Although, the authors have included the FPKM matrix in the supplementary, I strongly recommend the authors upload those information into a JBrowse on a public database such as <https://legumeinfo.org> so that readers can easily find out the expression level of the genes they are interesting in.

Reply: Many thanks for the suggestion. We have now made all the data/results reported in this study (transcriptome assembly, annotations and gene expression) publicly available for download via a web page (<https://ccbb.jnu.ac.in/CaGEA/>). The FPKM matrix file has also been revised to include the genomic location of all the transcripts. We plan to develop a comprehensive and user-friendly integrated database/browser for the chickpea gene expression data along with other datasets for the scientific community in near future.

2, The expression profile of the genes/transcripts in the QTL-hotspot is based on the ~2.8Mb region. A more recent paper has further narrowed down this to 113kb. The authors shall revise their analyses based on that. “Fine mapping studies identified a 113 kb region within “QTL-hotspot_a” for seed weight and drought related traits in chickpea”

Reply: Many thanks for the suggestion. We have now revised the analysis taking into consideration all the QTLs related to drought and salinity stresses identified in the previous studies and also integrated the DNA polymorphisms data to determine their influence on gene expression. It may be noted the reference suggested by the Reviewer belong to an abstract of a poster presentation. Nonetheless, the sub regions (QTL-hotspot_a and b) suggested have also been included in the revised analysis (lines 499-520).

3, The majority of the genes are highly expressed (several hundred-fold in some cases) in 20DAP tissue compared to other tissues and even 10 DAP, 30 DAP tissue closely related to 20 DAP. And almost one third of the tissue specific transcripts belong to 20DAP (2472/7903). It is important to discuss this further. Some genes without obvious function in floral, seed development such as nitrogen fixation (Ca_TC26389) were highly expressed in 20DAP. What could be the reason?

Reply: We have now added a few lines of discussion for these observations (lines 568-573).

Minor problems:

-L100 One advantage of the pacbio/IsoSeq is its ability to detect alternative splicing events, but it seems that the MS hasn't describe the Analysis of Alternative Splicing properly. Such as location and distribution of Alternative Splicing, GO term of genes having AS.

Reply: Many thanks for the comment. We have now incorporated the isoform level analysis including different types of alternative splicing events and their associated GO terms (lines 102-113; Figure S2).

L206-207 What is the cut-off of FPKM was used to declare “expressed”

Reply: A cut-off of >0.5 FPKM was set to identify expressed PCGs and lncRNAs and this has been mentioned in the MS now (line 198).

L413-415 The tissues in this study were not sampled under stress. I am surprised that a lot of stress responsive PCGs and lncRNAs were found in the root tissue.

Reply: Many thanks for the comment. A large number of genes involved in diverse biological processes have been implicated in stress responses. The representation of these

stress-responsive PCGs and lncRNAs in the root samples is not surprising, as their basal level of expression is also expected to be higher in the root tissues (which is the important tissue that is exposed to drought and salinity stresses at the first place).

L489 New paragraph starting from “The M6 and M7.”

Reply: Needful has been done.

L502 Why do the transcripts need to be tissue-specific expression? I assume there will be hormone signalling/responsive genes controlling 100 SDW, but they don't need to be specifically expressed in seed.

Reply: Many thanks for the comment. We agree that TFs and hormonal signalling components may play substantial role in trait determination including seed size/weight. However, the tissue-specific expression of certain genes provide an evidence of their implication in the biology of related tissues and may contribute in determining the identity of the tissues/organs including seed development and/or SDW determination to a large extent.

L505 TableS7 need references for the QTL

Reply: Needful has been done. The references have been provided in the additional Supplementary Tables (S7-S10) included in the revised manuscript.

-L807 Tissues from the same plant were used for Pacbio and illumina?

Reply: The same tissues were used for PacBio and Illumina sequencing. This has been clarified in the manuscript (lines 734-736).

-L866 The authors may need to justify using 10kb flanking region instead of other lengths

Reply: We used 10 kb flanking regions based on previous reports and the reference has been cited now (line 802).

-Fig9 Comparison of ER-Dtol-CT/Dsen-CT doesn't seem to have anything to do with drought stress. Can the authors provide Dtol-DS/Dsen-DS for ER, LR and Veg? Those are the drought responsive one

Reply: Many thanks for the comment. Although ER-Dtol-CT/Dsen-CT (for drought stress) doesn't seem to have anything to do with drought stress, it provides differences in the basal levels of gene expression between the two genotypes which may be important for their differential drought stress adaptation/response. Although we feel that a comparison of Dtol-DS/Dsen-DS may not provide direct evidence of their implication in drought stress response, we have analysed these comparisons too (Supplementary Data Set 6) and included comments on the comparative expression level of the set of genes identified in Dtol-DS/Dsen-DS and Stol-SS/Ssen-SS at ER, LR and Veg stages of development in the revised MS (lines 470-472; 491-493).

- In supplemental Data Set 1 &2, there is only Transcript id, however, it is also very important to show the chickpea gene id or position from the CDC Frontier v1.0 gene models as readers

are likely want to know the expression level of chickpea genes they are interested in. There are many transcriptome assemblies available in chickpea, few are available for visualisation in JBrowse due to lack of information or other reasons. It is really a waste of resource for the Chickpea community.

Reply: Many thanks for the comment. We have now added the genomic locations of all the transcripts in the FPKM matrix file. In addition, we have made all the data/results reported in this study (transcriptome assembly, annotations and gene expression) publicly available for download via a web page (<https://cbb.jnu.ac.in/CaGEA/>).

Reviewers' comments:

Reviewer #1 (Remarks to the Author):

The authors have made a significant effort to revise the document and I doing so they have addressed many of my concerns.

Nevertheless, I do have additional comments, as follows.

The data must be deposited into a public repository as a condition of publication, preferably at the NCBI. Making the data available on a server that the authors manage is entirely insufficient. Public databases have mandates to version data, which is essential for public reuse. Moreover, public databases are secure and immortal. No amount of guarantee from the authors can mitigate these concern.

Lines 207-208. GO term analysis identifies categories of annotation that may be differentially associated with process, in this case alternative splicing. However, the analysis generates "bins" of genes and is unable to identify explicit overlap of genes among biological processes. Thus, unless the authors have invested the effort to deconvolute the GO data so that the trajectories of individual genes can be followed, it is in appropriate to conclude that "the same set of genes may govern different biological processes via AS". I recommend removing this sentence.

It would be useful and interesting to understand the nature of the 6,981 genes that are missing from the genome assembly.

I personally find it distracting and unhelpful to use acronyms, except in certain common cases. For example, PCR for polymerase chain reaction, but not RTA for reference transcriptome assembly.

Lines 217-219. It is inappropriate to refer to the current data as "better" than prior data sets. More complete, certainly. Greater contiguity, apparently. But "better" is a subjective judgment -- one would need a comprehensive assessment of error rates to make this claim, which is not presented here.

Lines 220-222. I wonder about the veracity of the statement that "Of the 28,269 PCGs predicted in the chickpea genome, at least 25,221 were represented in our transcriptome with at least 90% identity and 30% coverage". 90% identity is very low considering that transcripts should share 100% identity (minus sequencing error rates) with the loci from which they originate. Indeed, there are numerous instances of genes that share considerably greater than 90% nucleotide identify. I think it is important that overlap between the transcriptome and genome be reported at 100% less expected error rates, or perhaps a graph depicting the number of genomic loci matching transcript isoforms at different levels of nucleotide identity.

Lines 223-225. Please divide the 13,597 transcripts into those not predicted in the existing chickpea genome annotation versus those not represented in the chickpea genome assembly.

Lines 298-299. The phrase "high specificity to the biological context" is ambiguous and I think it should be removed.

Lines 572-573. If one looks at Figure S11, there are 5 expression categories. The statement that most genes are either in the two top (FPKM >10) or two bottom (FPKM <5) is not surprising, since only one category remains. But more importantly, dividing genes into categories based on FPKM is arbitrary -- what is gained by doing so? If one wants to make a statement about the frequency distribution of gene expression within each tissue, then please make a graph showing this data without binning the genes into quintiles.

Lines 579-580. "Most of the samples were found clustered into groups according to their morphology (Figure 2b; Figure S9, S10)." The term "morphology" is non-sensical in this context. It might be correct to say that samples with similar developmental origins tended to be more similar, i.e., root and root derived samples, seeds samples, etc. But this is neither absolute, nor necessarily correct. The authors have drawn circles around samples with similar developmental origins, but these groupings are not statistically valid groups. According to PCA, two samples that are near one another have similar properties (composition and frequency) and as such there are many samples in one group that are actually more similar to samples in other groups. I am not disputing that these groupings are compelling, but rather that they reflect the authors' interpretation and preference, rather than statistically differentiated groups.

Regarding Figure 2B, a NMDS (Non-metric multidimensional scaling) is a more appropriate means to depict the data than PCA.

Line 580. "underground tissues" is not a biological category. Please re-word.

Lines 552-612. The analysis of DNA polymorphisms in transcripts near seed QTLs is misguided. QTLs are delimited by recombination -- they are haplotypes in this sense. With very few exceptions, only a single gene in a given QTL interval is causal to the trait phenotype. All of the others are simply along for the ride. Thus examining polymorphism rates in linked genes provides no insight about function. In fact it is misleading and invalid. Moreover, if one wanted to understand if polymorphism rates were greater in certain genome intervals, then one needs to understand genome-wide variation, using other more sophisticated metrics such as Tajima's D.

Reviewer #2 (Remarks to the Author):

The manuscript has been substantially improved. However, there are still some changes required, in my opinion:

-regarding the polymorphisms associated with the candidate genes underlying the QTLs, it seems important to me that the authors clarify the hypotheses that strengthen the roles of the candidates: 1. their localisation; 2. their expression; 3. their polymorphism: if a candidate gene does not show a polymorphism, how do the authors explain that it can play a determining role in the variation of the trait considered?

-The discussion remains difficult to read and needs to be largely rewritten.

L523-530: shorten sentences ; rephrase ;

L534-538: 3 times comprehensive ;

L534: what do you mean by high quality ; specify

L537: ...novel PCGs not annotated not represented in the chickpea genome, were ...

Line 542-544: reword

L548-595: shorten this paragraph;

L646-666: cf remark on causal polymorphisms

Reviewer #3 (Remarks to the Author):

I am satisfied with the authors' reply to my comments. One minor point is that it is not possible to download the data from the webpage (<http://ccbb.jnu.ac.in/CaGEA>). The authors might need to fix this before publication.

Reviewers' comments:

Reviewer #1 (Remarks to the Author):

The authors have made a significant effort to revise the document and I doing so they have addressed many of my concerns.

Reply: Many thanks for thoroughly reviewing the MS and for the positive comments.

Nevertheless, I do have additional comments, as follows.

The data must be deposited into a public repository as a condition of publication, preferably at the NCBI. Making the data available on a server that the authors manage is entirely insufficient. Public databases have mandates to version data, which is essential for public reuse. Moreover, public databases are secure and immortal. No amount of guarantee from the authors can mitigate these concern.

Reply: Many thanks for the comment. We agree with the Reviewer's suggestion. Please note that all the data had already been deposited to the public repository (including the transcriptome assembly) or made available via the Supplementary Files/Datasets in the manuscript. We have mentioned this more explicitly in the revised manuscript in the "Data availability" section.

Lines 107-108. GO term analysis identifies categories of annotation that may be differentially associated with process, in this case alternative splicing. However, the analysis generates "bins" of genes and is unable to identify explicit overlap of genes among biological processes. Thus, unless the authors have invested the effort to deconvolute the GO data so that the trajectories of individual genes can be followed, it is in appropriate to conclude that "the same set of genes may govern different biological processes via AS". I recommend removing this sentence.

Reply: Many thanks for the comment. The said sentence has been removed now, as suggested.

It would be useful and interesting to understand the nature of the 6,981 genes that are missing from the genome assembly.

Reply: Many thanks for the comment. We have now revised the analyses for this part and provided the list of genes (3603 transcripts) along with their annotation that did not map to the reference genome assembly in Supplemental Data Set 1.

I personally find it distracting and unhelpful to use acronyms, except in certain common cases. For example, PCR for polymerase chain reaction, but not RTA for reference transcriptome assembly.

Reply: Many thanks for the comment. The corrections at relevant places have been done.

Lines 217-219. It is inappropriate to refer to the current data as "better" than prior data sets. More complete, certainly. Greater contiguity, apparently. But "better" is a subjective judgment

-- one would need a comprehensive assessment of error rates to make this claim, which is not presented here.

Reply: Many thanks for the comment. The above-mentioned section has been moderated (removed the “better” word) in the revised MS, as suggested.

Lines 220-222. I wonder about the veracity of the statement that "Of the 28,269 PCGs predicted in the chickpea genome, at least 25,221 were represented in our transcriptome with at least 90% identity and 30% coverage". 90% identity is very low considering that transcripts should share 100% identity (minus sequencing error rates) with the loci from which they originate. Indeed, there are numerous instances of genes that share considerably greater than 90% nucleotide identity. I think it is important that overlap between the transcriptome and genome be reported at 100% less expected error rates, or perhaps a graph depicting the number of genomic loci matching transcript isoforms at different levels of nucleotide identity.

Reply: Many thanks for the comment. We have now performed the suggested analysis and added a bar graph depicting the number of genes showing different levels of identity (Figure S3) with the transcripts reported in the current study, as suggested by reviewer #1.

Lines 223-225. Please divide the 13,597 transcripts into those not predicted in the existing chickpea genome annotation versus those not represented in the chickpea genome assembly.

Reply: Many thanks for the comment. We have now revised the analysis and provided the list of transcripts (13586) along with their gene description in Supplemental Data Set 1, analyzed their functional annotation, and reported a summary of significantly enriched GO terms in the revised manuscript (Figure S8).

Lines 298-299. The phrase "high specificity to the biological context" is ambiguous and I think it should be removed.

Reply: Many thanks for the comment. The needful has been done.

Lines 572-573. If one looks at Figure S11, there are 5 expression categories. The statement that most genes are either in the two top (FPKM >10) or two bottom (FPKM <5) is not surprising, since only one category remains. But more importantly, dividing genes into categories based on FPKM is arbitrary -- what is gained by doing so? If one wants to make a statement about the frequency distribution of gene expression within each tissue, then please make a graph showing this data without binning the genes into quintiles.

Reply: Many thanks for the comment. We have now removed this data, as we find it is not providing any important information as also pointed out by the Reviewer.

Lines 579-580. "Most of the samples were found clustered into groups according to their morphology (Figure 2b; Figure S9, S10)." The term "morphology" is non-sensical in this context. It might be correct to say that samples with similar developmental origins tended to

be more similar, i.e., root and root derived samples, seeds samples, etc. But this is neither absolute, nor necessarily correct. The authors have drawn circles around samples with similar developmental origins, but these grouping are not statistically valid groups. According to PCA, two samples that are near one another have similar properties (composition and frequency) and as such there are many samples in one group that are actually more similar to samples in other groups. I am not disputing that these groupings are compelling, but rather that they reflect the authors' interpretation and preference, rather than statistically differentiated groups.

Reply: Many thanks for the comment. We revised the above-mentioned statements as suggested. The term “morphology” has been replaced with more appropriate terms, as suggested by the reviewer.

Regarding Figure 2B, a NMDS (Non-metric multidimensional scaling) is a more appropriate means to depict the data than PCA.

Reply: Many thanks for the comment. The correlation among transcriptomes of different tissues/organs has been reanalyzed using NMDS (Non-metric multidimensional scaling) and replaced with PCA in the revised MS, as suggested. It may be noted that the grouping of the tissues remains largely the same as was obtained via PCA in earlier analysis.

Line 580. "underground tissues" is not a biological category. Please re-word.

Reply: Many thanks for the comment. We revised the “underground tissues” as root-derived tissues in the revised MS.

Lines 552-612. The analysis of DNA polymorphisms in transcripts near seed QTLs is misguided. QTLs are delimited by recombination -- they are haplotypes in this sense. With very few exceptions, only a single gene in a given QTL interval is causal to the trait phenotype. All of the others are simply along for the ride. Thus examining polymorphism rates in linked genes provides no insight about function. In fact, it is misleading and invalid. Moreover, if one wanted to understand if polymorphism rates were greater in certain genome intervals, then one needs to understand genome-wide variation, using other more sophisticated metrics such as Tajima's D.

Reply: Many thanks for the comment. Although I agree with the Reviewer's comment that the analysis of DNA polymorphisms in transcripts located in the QTLs does not provide direct evidence about their function, we still believe that it can be helpful in the identification of a few better candidate genes for further functional validation. In fact, the analysis of expression profiles of the genes located within a candidate large genomic region is a standard method used for shortlisting the candidate gene/s causal for a phenotype followed by their/its functional validation.

We have analyzed the polymorphisms in genes within the QTL regions (reported earlier), as may be the better candidates for the agronomic traits. We correlated this analysis with the expression profiles to provide a clue about the putative role of these genes in important traits. The presence of DNA polymorphisms within the promoter region or cis-regulatory motifs can affect the binding of their cognate transcription factor thereby influencing the expression of downstream gene/s. Thus, the genes located in the QTL

region/s that harbor DNA polymorphism within their promoter region or cis-regulatory motifs are the better candidates for functional validation in future studies. I greatly appreciate the Reviewer's suggestion regarding the analysis of polymorphism rates in certain intervals as compared to the whole genome. However, this analysis is not a part of this study.

Reviewer #2 (Remarks to the Author):

The manuscript has been substantially improved. However, there are still some changes required, in my opinion:

-regarding the polymorphisms associated with the candidate genes underlying the QTLs, it seems important to me that the authors clarify the hypotheses that strengthen the roles of the candidates: 1. their localisation; 2. their expression; 3. their polymorphism: if a candidate gene does not show a polymorphism, how do the authors explain that it can play a determining role in the variation of the trait considered?

Reply: Many thanks for the positive comments. In many instances, QTLs are large genomic intervals that can accommodate numerous genes. To identify the putative genes located within these regions, we integrated the tissue-specificity and/or differential expression data along with the presence of DNA polymorphisms within the cis-regulatory elements. Although the genes located in the QTL regions showing tissue-specific/differential expression and harboring DNA polymorphisms within the cis-regulatory elements can provide important candidate genes, the plausible role of other genes in regulating the relevant biological processes for any trait cannot be ruled out. Those genes can also regulate the biological processes via interaction with other components or signaling pathways operative in different tissues or important traits.

-The discussion remains difficult to read and needs to be largely rewritten.

L523-530: shorten sentences ; rephrase ;

L534-538: 3 times comprehensive ;

L534: what do you mean by high quality ; specify

L537: ...novel PCGs not annotated not represented in the chickpea genome, were ...

Line 542-544: reword

L548-595: shorten this paragraph;

L646-666: cf remark on causal polymorphisms

Reply: Many thanks for the comments. The needful has been done to make the suggested corrections. In addition, we made a sincere effort to revise the statements in the discussion for better clarity.

Reviewer #3 (Remarks to the Author):

I am satisfy with the authors reply to my comments. One minor point is that it is not possible to download the data from the webpage (<http://ccbb.jnu.ac.in/CaGEA>). The authors might need to fix this before publication.

Reply: Many thanks for the favorable comment. The needful has been done. There was some network (LAN) issue on our campus for some time, which has been resolved now. The link is working properly now and we shall ensure its proper working in the future too. Nonetheless, as mentioned in the “Data availability” section, we have provided all the data/results from the manuscript in the public repository or as Supplemental files/datasets in the manuscript itself.

REVIEWERS' COMMENTS:

Reviewer #1 (Remarks to the Author):

The authors have done a commendable job of responding to my previous concerns. Thank you for that.

I have one major and one minor issue that remain before I consider the revisions to be complete.

First, and most importantly, the data **MUST** be deposited to an independent data repository that is intended specifically for genomics data. The most obvious repository is NCBI.

The server at JNU is unacceptable.

Please keep in mind that repositories such as NCBI have the very significant advantages of security, versioning and interconnectedness with the majority of the scientific community's genomic data. Data at NCBI are accessible to users in a common background of query tools and analyses that are unmatched in their sophistication and that greatly increase the data's value.

Minor issues:

The acronym DAP is used throughout the document, but it is never spelled out.

Reviewer #2 (Remarks to the Author):

The authors have addressed my questions. I thus consider that the manuscript can be published in Nat Comm. However, I still have some comments on the form of the MS :

L94-97: not clear

L159: lncRNAs represent a non-coding component > there are other non coding component in the genome !

L163: indicate where this data can be found ? which table ?

L186: Person correlations : is it r or r2 ?

L204: podshell > (part of the pod is an awkward definition !

L475: harboured DNA polymorphisms

L509: previous chickpea transcriptomes

L511: ...availability of the long-read PacBio technology now allows the generation...

L512: ...error-rate of this technology and the cost of sequencing large ...

L513: ...the representation of even lowly expressed transcripts, it is advisable ...

L516: Here, we report ...

L518: in set after "biological replicates" the sentence L520-521: A significant ...assembly". Followed by: "We also annotated a set of ..."

L522-525: move sentence to next §; Give the web address of you gene expression atlas;

L533-534 redundant with L522-525.

L540-542: delete

L543: "more subtle specific expression patterns" ... what does this mean?

P18-19-20-21: it is still too long with some paragraphs that are redundant with the results section; I think it could be reduced to 2/3;

Reviewers' comments:

Reviewer #1 (Remarks to the Author):

The authors have done a commendable job of responding to my previous concerns. Thank you for that.

Reply: Many thanks for the encouraging comments.

I have one major and one minor issue that remain before I consider the revisions to be complete.

First, and most importantly, the data **MUST** be deposited to an independent data repository that is intended specifically for genomics data. The most obvious repository is NCBI.

The server at JNU is unacceptable.

Please keep in mind that repositories such as NCBI have the very significant advantages of security, versioning and interconnectedness with the majority of the scientific community's genomic data. Data at NCBI are accessible to users in a common background of query tools and analyses that are unmatched in their sophistication and that greatly increase the data's value.

Reply: Many thanks for the valuable suggestion and we totally agree with the Reviewer's concern. However, we have already deposited the raw and the associated supplementary files in the public repositories (SRA and GEO) at NCBI. This is clearly mentioned in the manuscript. Moreover, all the relevant analyzed files are available as Supplementary Tables and Data along with the manuscript. In addition, we have made the data available via the webpage <http://cbb.jnu.ac.in/CaGEA> at JNU server.

Minor issues:

The acronym DAP is used throughout the document, but it is never spelled out.

Reply: Many thanks for the suggestion. We have included the expansion of DAP as “days after pollination” at the first appearance.

Reviewer #2 (Remarks to the Author):

The authors have addressed my questions. I thus consider that the manuscript can be published in Nat Comm. However, I still have some comments on the form of the MS :

Reply: Many thanks the comprehensive review of the MS and encouraging comment.

L94-97: not clear

Reply: We have revised the sentences for better clarity.

L159: lncRNAs represent a non-coding component > there are other non-coding component in the genome !

Reply: Many thanks for the suggestion. We have moderated the sentence now.

L163: indicate where this data can be found ? which table ?

Reply: We have cited the relevant Supplementary Data file in the revised MS.

L186: Person correlations : is it r or r² ?

Reply: We have included the type of Pearson correlation (r) in the revised MS, as suggested.

L204: podshell > (part of the pod is an awkward definition !

Reply: Many thanks for the suggestion. We have removed this definition.

L475: harboured DNA polymorphisms

Reply: Correction has been done, as suggested.

L509: previous chickpea transcriptomes

Reply: Correction has been done, as suggested.

L511: ...availability of the long-read PacBio technology now allows the generation...

Reply: Correction has been done, as suggested.

L512: ...error-rate of this technology and the cost of sequencing large ...

Reply: Correction has been done, as suggested.

L513: ...the representation of even lowly expressed transcripts, it is advisable ...

Reply: Correction has been done, as suggested.

L516: Here, we report ...

Reply: Correction has been done, as suggested.

L518: in set after “biological replicates” the sentence L520-521: A significant ...assembly”. Followed by: “We also annotated a set of ...”

Reply: Many thanks for the suggestion. The needful has been done.

L522-525: move sentence to next §; Give the web address of you gene expression atlas;

Reply: Many thanks for the suggestion. The needful has been done.

L533-534 redundant with L522-525.

Reply: Many thanks for the suggestion. The redundancy has been removed.

L540-542: delete

Reply: We deleted the sentence as suggested.

L543: “more subtle specific expression patterns” ... what does this mean?

Reply: We have now revised the sentence for more clarity.

P18-19-20-21: it is still too long with some paragraphs that are redundant with the results section; I think it could be reduced to 2/3;

Reply: Many thanks for the suggestion. We have made sincere efforts to remove redundant statements and concise the paragraphs at pages 18-21, as suggested.